# MIR-BFT: SCALABLE AND ROBUST BFT FOR DECENTRALIZED NETWORKS

CHRYSOULA STATHAKOPOULOU ⦿
*IBM Research Europe - Zurich*
*ETH Zurich*

TUDOR DAVID
*Oracle Labs* *

MATEJ PAVLOVIC ⦿
*IBM Research Europe - Zurich*

MARKO VUKOLIĆ
*Protocol Labs* *

## Abstract

This paper presents Mir-BFT, a robust Byzantine fault-tolerant (BFT) total order broadcast protocol aimed at maximizing throughput on wide-area networks (WANs), targeting deployments in decentralized networks, such as permissioned and Proof-of-Stake permissionless blockchain systems.

Mir-BFT is the first BFT protocol that allows multiple leaders to propose request batches independently (i.e., parallel leaders), while effectively precluding performance degradation due to request duplication by rotating the assignment of a partitioned request hash space to leaders. As this mechanism removes the single-leader bandwidth bottleneck and exposes a computation bottleneck related to authenticating clients even on a WAN, our protocol further boosts throughput using a client signature verification sharding optimization. Our evaluation shows that Mir-BFT outperforms state-of-the-art single-leader protocols and orders more than 60000 signed Bitcoin-sized (500-byte) transactions per second on a widely distributed setup (100 nodes, 1 Gbps WAN) with typical latencies of few seconds. Moreover, our evaluation exposes the impact of duplicate requests on parallel leader protocols which Mir-BFT eliminates. We also evaluate Mir-BFT under different crash and Byzantine faults, demonstrating its performance robustness.

Mir-BFT relies on classical BFT protocol constructs, which simplifies reasoning about its correctness. Specifically, Mir-BFT is a generalization of the celebrated and scrutinized PBFT protocol. In a nutshell, Mir-BFT follows PBFT "safety-wise", with changes needed to accommodate novel features restricted to PBFT liveness.

## 1 Introduction

**Background.** Byzantine fault-tolerant (BFT) protocols, which tolerate malicious (Byzantine [47]) behavior of a subset of nodes, have evolved from being a niche technology for tolerating bugs and intrusions to be the key technology to ensure consistency of widely deployed decentralized networks in which multiple mutually untrusted parties administer different nodes (such as in blockchain systems) [27, 35, 58]. Specifically, BFT protocols are considered to be an alternative to (or complementing) energy-intensive and slow Proof-of-Work (PoW) consensus protocols used in early blockchains including Bitcoin [34, 58]. BFT protocols relevant to decentralized networks are consensus and total order (TO) broadcast protocols [21] which establish the basis for state-machine replication (SMR) [54] and smart-contract execution [60].

BFT protocols are known to be very efficient on small scales (few nodes) in clusters (e.g., [13, 44]), or to exhibit modest performance on large scales (thousands or more nodes) across wide area networks (WAN) (e.g., [35]). Recently, considerable research effort (e.g., [19, 26, 36, 51, 61]) focused on maximizing BFT performance in medium-sized WAN networks (on the order of 100 nodes), as this deployment setting is highly relevant to different types of decentralized networks.

On the one hand, *permissioned* blockchains, such as Hyperledger Fabric [11], are rarely deployed on scales above 100 nodes. Yet use cases gathering dozens of organizations, which do not necessarily trust each other, are very prominent [2]. On the other hand, this setting is also highly relevant in the context of large scale *permissionless* blockchains, in which anyone can participate, that use weighted voting (based, e.g., on Proof-of-Stake (PoS) [20, 40] or delegated PoS (DPoS) [5]) or committee-voting [35] to limit the number of nodes involved in the critical path of the consensus protocol. With such weighted voting the number of (relevant) nodes for PoS/DPoS consensus is typically on the order of a hundred [5] or sometimes even less [8]. Related open-membership blockchain systems, such as Stellar, also run consensus among less than 100 nodes [48].

**Challenges.** Most of this research (e.g., [19, 26, 36, 61]) aims at addressing the scalability issues that arise in classical leader-based BFT protocols, such as the seminal PBFT protocol [24]. In short, in a leader-based proto-

---

*Work done while at IBM Research Europe - Zurich

col, a leader, who is tasked with assembling a batch of requests (or block of transactions) and communicating it to all other nodes, has at least $O(n)$ work, where $n$ is the total number of nodes. Hence the leader quickly becomes a bottleneck as $n$ grows.

A promising approach to addressing scalability issues in BFT is to allow multiple nodes to act as *parallel leaders* and to propose batches independently and concurrently either in a coordinated, deterministic fashion [26,38,52] or using randomized protocols [28,45,51]. With parallel leaders, the CPU and bandwidth load related to proposing batches are distributed more evenly. However, the issue with this approach is that parallel leaders are prone to wasting resources by proposing the same duplicate requests. As depicted in Table 1, none of the current BFT protocols that allow for parallel leaders deal with request duplication, which is straightforward to satisfy in single leader protocols. The tension between preventing request duplication and using parallel leaders stems from two important attacks that an adversary can mount and an efficient BFT protocol needs to prevent: (i) the *request censoring attack* by Byzantine leader(s), in which a malicious leader simply drops or delays a client's request (transaction), and (ii) the *request duplication attack*, in which Byzantine clients submit the exact same request multiple times.

To counteract request censoring attacks, a BFT protocol needs to allow at least $f + 1$ different leaders to propose a request (where $f$, which is typically $O(n)$, is the threshold on the number of Byzantine nodes in the system). Single-leader protocols (e.g., [24,61]), which typically rotate the leadership role across all nodes, address duplication attacks relatively easily. On changing the leader, a new leader only needs to make sure they do not repeat requests previously proposed by previous leaders.

With parallel leaders, the picture changes substantially. If a (malicious or correct) client submits the same request to multiple parallel leaders concurrently, the parallel leaders will include the same request in their respective batches, i.e., they will order duplicates of the same request. While these duplicates can simply be filtered out after ordering (or after the reception of a duplicate, during ordering), the damage has already been done — excessive resources, bandwidth and possibly CPU have been consumed. To complicate the picture, naïve solutions in which: (i) clients are requested to sequentially send to one leader at the time, (ii) the leader randomly samples a queue of pending requests, or (iii) clients pay transaction fees for each duplicate, do not help.

In the first case, Byzantine clients mounting a request

|  | Parallel Leaders | Prevents Req. Duplication |
|---|---|---|
| PBFT [24] | no | yes |
| BFT-SMaRt [17] | no | yes |
| Aardvark [25] | no | yes |
| RBFT [12] | no | yes |
| Spinning [57] | no | yes |
| Prime [10] | no | yes |
| 700 [13] | no | yes |
| Zyzzyva [44] | no | yes |
| SBFT [36] | no | yes |
| HotStuff [61] | no | no[1] |
| Tendermint [19] | no | yes |
| BFT-Mencius [52] | **yes** | no |
| RedBelly [26] | **yes** | no |
| RCC [38] | **yes** | no[2] |
| OMADA [30] | **yes** | no |
| Hashgraph [45] | **yes** | no |
| Honeybadger [51] | **yes** | no |
| BEAT [28] | **yes** | no |
| **Mir (this paper)** | **yes** | **yes** |

Table 1: Comparison of Mir to related BFT protocols.
[1] duplication could easily be prevented.
[2] duplication can be prevented under stronger synchrony assumptions.

duplication attack are not required to respect sending a request sequentially. Moreover, such a behavior cannot be distinguished from a correct client who simply sends a transaction multiple times due to asynchrony or network issues.

Random sampling of the pending requests (Honeybadger [51], BEAT [28]) proves ineffective in practice. When a node constructs a new proposal from randomly chosen pending (i.e., received and yet unproposed) requests, its proposal might still intersect with that of another node. This is especially likely if the system is not in deep saturation and nodes' buffers of pending requests are small.

The third case concerns some blockchain systems, such as Hedera Hashgraph, which charge transaction fees for every duplicate request [15]. This approach, however, penalizes correct clients when they resubmit a transaction to counteract possible censoring attacks, or a slow network. In more established decentralized systems, such as Bitcoin and Ethereum, it is standard to charge for the same transaction only once, even if it is submitted by a client more than once.

In summary, with up to $O(n)$ parallel leaders, request duplication attacks may induce an $O(n)$-fold duplication of every single request and bring the effective throughput

to its knees, practically voiding the benefits of using multiple leaders.

**Contributions.** This paper presents Mir-BFT, (or, simply, Mir) [1]), a novel BFT total order broadcast (TOB) protocol which is the first to combine parallel leaders with robustness to request duplication. Mir also addresses notable performance attacks [25], such as the Byzantine leader straggler attack. Mir is further robust to arbitrarily long, yet finite, periods of asynchrony and is optimally resilient (requiring optimal $n \geq 3f + 1$ nodes to tolerate $f$ Byzantine faulty ones). On the performance side, Mir achieves the best throughput, when compared to legacy and state-of-the-art TOB protocols, on public WAN networks, as confirmed by our measurements on up to 100 nodes. The following summarizes the main features of Mir, as well as contributions of this paper:

• Mir allows multiple parallel leaders to propose batches of requests concurrently, in a sense multiplexing several PBFT instances into a single total order, in a robust way. As its main novelty, Mir partitions the request hash space and distributes its subsets to the leaders, preventing request duplication. To also prevent censoring attacks Mir periodically re-distributes this partitioned assignment to the leaders.

• Mir further uses a *client signature verification sharding* throughput optimization to offload CPU, which is exposed as a bottleneck in Mir once we remove the single-leader bandwidth bottleneck.

• Mir avoids "design-from-scratch", which is known to be error-prone for BFT [9, 13]. Mir is a generalization of the well-scrutinized PBFT protocol [2], which it closely follows "safety-wise" while introducing important generalizations only affecting PBFT liveness (e.g., (multiple) leader election). This simplifies the reasoning about Mir's correctness.

• We implement Mir in Go and run it with up to 100 nodes in a multi-datacenter WAN, as well as in clusters, and under different faults, comparing it to state of the art BFT protocols. Our results show that Mir convincingly outperforms state of the art, ordering more than 60000 signed Bitcoin-sized (500-byte) requests per second (req/s), with typical latencies of few seconds. In this setup, Mir achieves 3x the throughput of the optimistic sub-protocol of Aliph [13], Chain, and more than an order of magnitude higher throughput than other state of the art single-leader BFT protocols. To put this into

---

[1] In a number of Slavic languages, the word *mir* refers to universally good, global concepts, such as *peace* and/or *world*.

[2] Mir variants based on other BFT protocols can be derived as well.

perspective, Mir's 60000+ req/s on 100 nodes on WAN is 2.5x the alleged peak capacity of VISA (24k req/s [6]) and more than 30x faster than the actual average VISA transaction rate (about 2k req/s [58]).

**Roadmap.** The rest of the paper is organized as follows. In Section 2, we define the system model and in Section 3 we briefly present PBFT (for completeness). In Section 4, we give an overview of Mir and changes it introduces to PBFT. We then explain Mir implementation details in Section 5. We further list the Mir's pseudocode in Section 6.

This is followed by Mir's correctness proof in Section 7. Section 8 introduces an optimization tailored to large requests, such as the ones featured by Hyperledger Fabric. Section 9 gives evaluation details. Finally, Section 10 discusses related work and Section 11 concludes.

## 2  System Model

We assume an eventually synchronous system [29] in which the communication among *correct* processes can be fully asynchronous before some global synchronization time (*GST*), unknown to nodes, after which it is assumed to be synchronous. Processes are split into a set of *n nodes*, denoted by *Nodes*, and a set of *clients*. We assume a public key infrastructure in which processes are identified by their public keys; we further assume that node identities are lexicographically ordered and mapped by a bijection to the set $[0 \dots n-1]$ which we use to reason about node identities. In every execution, at most $f$ nodes can be *Byzantine* faulty (i.e., crash or deviate from the protocol in an arbitrary way), such that $n \geq 3f + 1$. Any number of clients can be Byzantine.

We assume an adversary that can control Byzantine faulty nodes but cannot break the cryptographic primitives we use, such as PKI and cryptographic hashes (we use SHA-256). $H(data)$ denotes a cryptographic hash of *data*, while $data_{\sigma_p}$ denotes *data* signed by process $p$ (client or node). Processes communicate through authenticated point-to-point channels (our implementation uses gRPC [4] over TLS, preventing man-in-the-middle and related attacks).

Nodes implement a BFT total order (atomic) broadcast service to clients. To broadcast request $r$, a client invokes BCAST($r$). A client request is a tuple $r = (o, t, c)$, where $o$ is the request payload, e.g., some operation to be executed by some application, $c$ is a unique client identifier, e.g., the client's public key, and $t$ is the client timestamp. $t$ is a *logical* timestamp, effectively count-

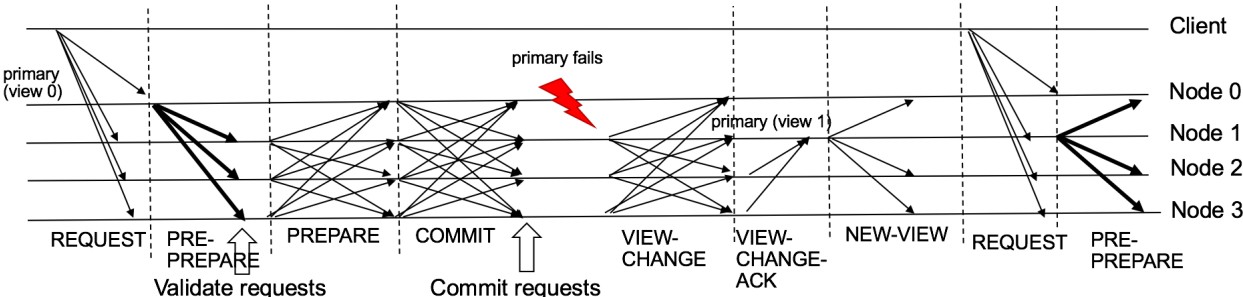

Figure 1: PBFT communication pattern and messages. Bottleneck messages are shown in **bold**.

ing the requests submitted by client $c$. The request is wrapped in a message $\langle \text{REQUEST}, r \rangle_{\sigma_c}$.

Two client requests $r = (o,t,c), r' = (o',t',c')$ are considered the same, we write $r = r'$ and we refer to them as duplicates, if and only if $o = o' \wedge t = t' \wedge c = c'$.

Nodes eventually output $\text{DELIVER}(sn,r)$, such that the following properties hold:

P1 **Validity:** If a correct node delivers $r$, then some client broadcast $r$.

P2 **Agreement (Total Order):** If two correct nodes deliver requests $r$ and $r'$ with sequence number $sn$, then $r = r'$.

P3 **No duplication:** If a correct node delivers request $r$ with sequence numbers $sn$ and $sn'$, then $sn = sn'$.

P4 **Totality:** If a correct node delivers request $r$, then every correct node eventually delivers $r$.

P5 **Liveness:** If a correct client broadcasts request $r$, then some correct node $p$ eventually delivers $r$.

P6 **In-order delivery:** If a correct $i$ node delivers some request $r$ with sequence number $sn$, then $i$ has delivered requests for each sequence number $sn'$, such that $sn' < sn$.

Note that P3 (No duplication) is a standard TOB property [21] that most protocols can easily satisfy by filtering out duplicates *after* agreeing on request order, which is bandwidth wasting. Mir enforces P3 *without ordering duplicates*, using a novel approach to eliminate duplicates during agreement to improve performance and scalability. Notice that a client can still broadcast multiple requests with the same payload, which might be meaningful with respect to the semantics of some applications. Those requests should, however, have different client timestamps.

Client timestamps, therefore, allow such "intended" duplication and distinguish it from "accidental" or Byzantine request duplication.

Property P6 (In-order delivery) is typically not required to guarantee TOB. However, it allows an external application to execute the requests with the same order in which they are delivered. Importantly, Mir does not execute requests. Request execution is orthogonal and can be performed in any execution model, such as the order-execute model implemented in systems like Ethereum or the execute-order-validate model implemented in systems like Fabric.

Notice, moreover, that, because of properties P3 and P6, request sequence numbers do not provide any additional information. Each request is only delivered once and in the same order by each node in *Nodes*. Therefore, $\text{DELIVER}(sn,r)$ is equivalent to $\text{DELIVER}(r)$ (the sequence number $sn$ being implied by the number of requests delivered before $r$). We keep, however, the explicit sequence number in the interface of our system to ease the reader and facilitate compatibility with external applications which index requests with some sequence number.

## 3 PBFT and its Bottlenecks

We depict the PBFT communication pattern in Figure 1. PBFT proceeds in rounds called *views* which are led by the *primary*. The primary sequences and *proposes* a client's request (or a batch thereof) in a PRE-PREPARE message — on WANs this step is typically a network bottleneck. Upon reception of the PRE-PREPARE, other nodes validate the request, which involves, at least, verifying its authenticity (we say nodes *preprepare* the request). This is followed by two rounds of all-to-all communication (PREPARE and COMMIT messages), which are not bottlenecks as they leverage $n$ links in parallel and

| Protocol | PBFT [24] | Mir |
|---|---|---|
| Client request authentication | vector of MACs (1 for each node) | signatures |
| Batching | no (or, 1 request per "batch") | yes |
| Multiple-batches in parallel | yes (watermarks) | yes (watermarks) |
| In-order delivery | no | yes |
| Round structure/naming | views | epochs |
| Round-change responsibility | view primary (round-robin across all nodes) | epoch primary (round-robin across all nodes) |
| No. of per-round leaders | 1 (view primary) | many (from 1 to $n$ epoch leaders) |
| No. of batches per round | unbounded | bounded (*ephemeral* epochs); unbounded (*stable* epochs) |
| Round leader selection | primary is the only leader | primary decides on epoch leaders (subject to constraints) |
| Request duplication prevention | enforced by the primary | hash space partitioning across epoch leaders |

Table 2: High level overview of the original PBFT [24] vs. Mir protocol structure.

contain metadata (request/batch hash) only. A node *prepares* a request and sends a COMMIT message if it gets a PREPARE message from a quorum ($n - f \geq 2f + 1$ nodes) that matches a PRE-PREPARE. Finally, nodes *commit* the request in total order, if they get a quorum of matching COMMIT messages.

The primary is changed only if it is faulty or if asynchrony breaks the availability of a quorum. In this case, nodes timeout and initiate a *view change*. View change involves communication among nodes in which they exchange information about the latest *preprepared* and *prepared* requests, such that the new primary, which is selected in a round-robin fashion, must re-propose potentially committed requests under the same sequence numbers within a NEW-VIEW message (see [24] for details). The view-change pattern can be simplified using signatures [23].

After the primary is changed, the system enters the new view and common-case operation resumes. PBFT complements this main common-case/view-change protocols with *checkpointing* (log and state compaction) and state transfer sub-protocols [24].

## 4 Mir Overview

Mir is based on PBFT [24] (Sec. 3). In a nutshell, Mir executes multiple instances of PBFT in parallel, in which all nodes participate. Nodes *commit* a batch with a sequence number upon receiving a quorum of matching COMMIT messages, same as in PBFT. Finally, nodes *deliver* the requests of a batch, or, for short, deliver the batch, once they have committed it with some sequence number and they have delivered all batches with smaller sequence numbers. Major differences between PBFT and Mir are summarized in Table 2. In this section we elaborate on these differences, giving a high-level overview of Mir.

**Request Authentication.** While PBFT authenticates clients' requests with a vector of MACs, Mir uses signatures for request authentication to avoid concerns associated with "faulty client" attacks related to the MAC authenticators, which PBFT uses, [25] and to prevent any number of colluding nodes, beyond $f$, from impersonating a client. However, this change may induce a throughput bottleneck, as per-request verification of clients' signatures requires more CPU than that of MACs. We address this issue by a signature verification sharding optimization described in Sec. 5.6.

**Batching and Watermarks.** Mir processes requests in *batches* (ordered lists of requests formed by a leader), a standard throughput improvement of PBFT (see e.g., [13, 44]). Mir also retains the request/batch *watermarks* used by PBFT to boost throughput. In PBFT, request watermarks, low and high, represent the range of request sequence numbers which the primary/leader can propose concurrently. While many successor BFT protocols eliminated watermarks in favor of batching (e.g, [13, 17, 44]), Mir reuses watermarks to facilitate concurrent proposals of batches by *multiple parallel leaders*.

**Protocol Round Structure.** Unlike PBFT, Mir distinguishes between leaders and a primary node. Mir proceeds in *epochs* which correspond to *views* in PBFT, each epoch having a single *epoch primary* — a node deterministically defined by the epoch number, by round-robin rotation across all the participating nodes of the protocol.

Each epoch $e$ has a set of *epoch leaders* (denoted by $EL(e)$), which we define as nodes that can sequence and propose batches in $e$. In contrast, in PBFT, only the

primary is a leader. Within an epoch, Mir deterministically partitions sequence numbers across epoch leaders, such that all leaders can propose their batches simultaneously without conflicts. Epoch $e$ transitions to epoch $e+1$ if (1) one of the leaders is suspected of failing, triggering a timeout at sufficiently many nodes (*ungracious epoch change*), or (2) a predefined number of batches $maxLen(e)$ has been delivered (*gracious epoch change*). While the ungracious epoch change corresponds exactly to PBFT's view change, the gracious epoch change is a much more lightweight protocol.

**Selecting Epoch Leaders.** For each epoch, it is the *primary* who *selects the leaders* and reliably broadcasts its selection to all nodes. In principle, the primary can pick an arbitrary leaderset as long as the primary itself is included in it. We evaluate a simple "grow on gracious, reduce on ungracious epoch" policy for leaderset size. If $i$ starts epoch $e$ with an ungracious epoch change and $e'$ is the last epoch for which $i$ knows the epoch configuration, $i$ adds itself to the leaderset of epoch $e'$ and removes one node (not itself) for each epoch between $e$ and $e'$ (leaving at least itself in the leaderset). If the epoch change occurred from the expiration of *ecTimer* on some sequence number *sn*, the next epoch primary chooses to remove the node to whom *sn* was assigned.

Moreover, in an epoch $e$ where all nodes are leaders ($EL(e) = Nodes$), we define $maxLen(e) = \infty$ (i.e., $e$ only ends if a leader is suspected). Otherwise, $maxlen(e)$ is a constant, pre-configured system parameter. We call the former *stable* epochs and the latter *ephemeral*.

More elaborate strategies for choosing epoch lengths and leadersets, which are outside the scope of this paper, can take into account execution history, fault patterns, weighted voting, distributed randomness, or blockchain stake. Note that with a policy that constrains the leaderset to only the epoch primary and makes every epoch stable, Mir reduces to PBFT.

**Request Duplication and Request Censoring Attacks.** Moving from single-leader PBFT to multi-leader Mir poses the challenge of request duplication. A simplistic approach to multiple leaders would be to allow any leader to add any request into a batch ( [26, 45, 52]), either in the common case, or in the case of client request retransmission. Such a simplistic approach, combined with a client sending a request to exactly one node, allows good throughput with no duplication only in the best case, i.e., with no Byzantine clients/leaders and with no asynchrony.

However, this approach does not perform well outside the best case, in particular with clients sending identical requests to multiple nodes. A client may do so simply because it is Byzantine and performs the *request duplication* attack. However, even a correct client needs to send its request to at least $f + 1$ nodes (i.e., to $\Theta(n)$ nodes, when $n = 3f + 1$) in the worst case in *any* BFT protocol, in order to avoid Byzantine nodes (leaders) selectively ignoring the request (*request censoring* attack). Therefore, a simplistic approach to parallel request processing with multiple leaders [26, 45, 52] faces attacks that can reduce throughput by factor of $\Theta(n)$, nullifying the effects of using multiple leaders.

Note the subtle but important difference between a duplication attack (submitting the same request to multiple replicas) and a DoS attack (submitting many different requests) that a Byzantine client can mount. A system can prevent the latter (DoS) by imposing per-client limits on the incoming *unique* request rate. Mir enforces such a limit through client request watermarks. A duplication attack, however, is resistant to such mechanisms, as a Byzantine client is indistinguishable from a correct client with a less reliable network connection. We demonstrate the effects of these attacks in Section 9.5.

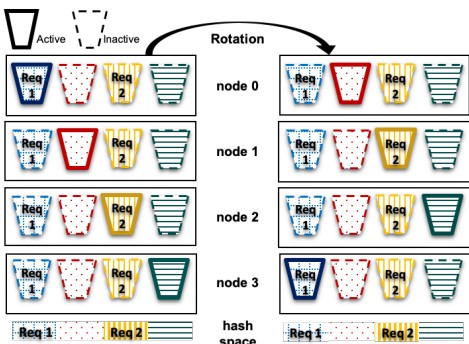

Figure 2: Request mapping in a stable epoch with $n = 4$ (all nodes are leaders): Solid lines represent the active buckets. Req. 1 is mapped to the first bucket, first active in node 1. Req. 2 is mapped to the third bucket, first active in node 3. Rotation redistributes bucket assignment across leaders.

**Buckets and Request Partitioning.** To cope with these attacks, Mir partitions the request hash space into buckets of equal size (number of buckets is a system parameter) and assigns each bucket to exactly one leader, allowing a leader to only propose requests from its as-

signed (*active*) buckets (preventing request duplication). For load balancing, Mir distributes buckets evenly (within the limits of integer arithmetics) to all leaders in each epoch. To prevent request censoring, Mir makes sure that every bucket will be assigned to a correct leader infinitely often. We achieve this by periodically redistributing the bucket assignment. Bucket re-distribution happens (1) at each epoch change (see Sec. 5.2) and (2) after a predefined number of batches have been delivered in a stable epoch (since a stable epoch might never end), as illustrated in Figure 2. Note that all nodes, while proposing only requests from their active buckets, still receive and store all requests (this can be optimized, see 5.1).

**Parallelism.** The Mir implementation (detailed in Sec. 5.10) is highly parallelized, with every *worker* thread responsible for one batch. In addition, Mir uses multiple gRPC connections among each pair of nodes which proves to be critical in boosting throughput in a WAN especially with a small number of nodes.

**Generalization of PBFT and Emulation of Other BFT Protocols.** Mir can be easily configured to implement or approximate single leader protocols. In particular, in PBFT, each epoch has a single leader, the primary, same for all batches, and an epoch change occurs only when the primary is suspected to be faulty. To reduce Mir to PBFT we simply enforce a single leader in each epoch, the primary node, and an infinite number of sequence numbers for each epoch; thus all epochs are stable and all epoch changes ungracious. This results in the signle epoch leader being responsible for all buckets, and, therefore, hides the bucket re-distribution sub-protocol within a stable epoch. Other protocols, such as Tendermint [19] and Spinning [57], rotate the leader per sequence number. To approximate such protocols, we fix the maximum number of sequence numbers and leaders in every epoch to 1. This results in a gracious epoch change per sequence number and therefore rotating the leaders (the epoch primary) with every batch.

# 5   Mir Implementation Details

## 5.1   The Client

Upon BCAST($r$), i.e., broadcasting a request $r$, a client $c$ creates a message $\langle \text{REQUEST}, r \rangle_{\sigma_c}$, where $r = (o, t, c)$ is a payload, timestamp, public key tuple as described in Section 2. The client timestamp $t$, must be in a sliding window between the low and high *client watermark*

$t_{c_L} < t \leq t_{c_H}$. Client watermarks in Mir allow multiple requests originating from the same client to be "in-flight", to enable high throughput without excessive number of clients. These watermarks are periodically advanced with the checkpoint mechanism described in Section 5.5, in a way which leaves no unused timestamps. Mir alligns checkpointing/advancing of the watermarks with bucket re-distributions (when no requests are in flight), such that all nodes always have a consistent view of the watermarks.

In principle, the client sends the REQUEST to all nodes (and periodically re-sends it to those nodes who have not received it, until the request is delivered by at least $f + 1$ nodes). In practice, a client may start by sending its request to fewer than $n$ nodes ($f + 1$ in our implementation) and only send it to the remaining nodes if the request has not been delivered by $f + 1$ nodes after a timeout.

## 5.2   Sequence Numbers and Buckets

**Sequence Numbers.** In each epoch $e$, a leader may only use a subset of $e$'s sequence numbers for proposing batches. Mir partitions $e$'s sequence numbers to leaders in $EL(e)$ in a round-robin way, using modulo arithmetic, starting at the epoch primary (see Fig. 3 and Alg. 4, Line 179). We say that a leader *leads* sequence number $sn$ when the leader is assigned $sn$ and is thus expected to send a PRE-PREPARE for the batch with sequence number $sn$. Batches are proposed in parallel by all epoch leaders and are processed like in PBFT. Recall (from Table 2) that batch watermarking (not to be confused with client request watermarking from Sec. 5.1) allows the PBFT primary to propose multiple batches in parallel; in Mir, we simply extend this to multiple leaders.

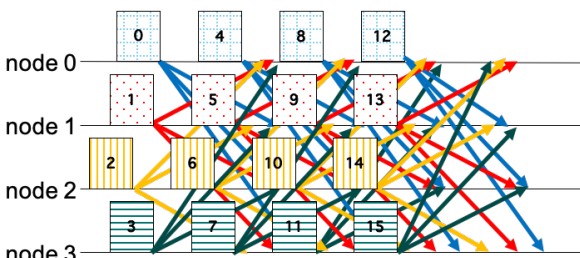

Figure 3: PRE-PREPARE messages in an epoch where all 4 nodes are leaders balancing the proposal load. Mir partitions batch sequence numbers among epoch leaders.

**Buckets.** In epoch $e = 0$, we assign buckets to leaders sequentially. Recall, with the term bucket we refer to a subset of the request hash space. We start the assignment from the buckets with the lowest hash values which we assign to the primary of epoch 0. For $e > 0$, the primary picks a set of consecutive buckets for itself (primary's *preferred buckets*), starting from the bucket which contains the *oldest* request it received; this is key to ensuring Liveness (P5, Sec. 2). Mir distributes the remaining buckets evenly and deterministically among the other leaders — this distribution is determined from an *epoch configuration* which the epoch primary reliably broadcasts and which contains preferred buckets and leaderset selection (see Sec. 5.4.2). Buckets assigned to a leader are called its *active* buckets.

Additionally, if $e$ is stable (when $maxLen(e) = \infty$ and thus no further epoch changes are guaranteed), leaders periodically rotate the bucket assignment (each time a pre-configured number of batches are delivered): leader $i$ is assigned buckets previously assigned to leader $i + 1$ (in modulo $n$ arithmetic). To prevent accidental request duplication, which could result in leader $i$ being suspected and removed from the leaderset, leader $i$ waits to deliver all "in-flight" batches before starting to propose its own batches (Alg. 4, Line 68 and Alg. 4, Lines 194-198). Other nodes do the same before prepreparing batches in $i$'s new buckets. In the example shown in Fig. 2, after the bucket re-distribution (rotation), node 0 waits to deliver all batches (still proposed by node 1) from its newly active red (second) bucket, before node 0 starts proposing new batches from the red (second) bucket.

## 5.3 Common Case Operation

**REQUEST.** In the common case, the protocol proceeds as follows. Upon receiving $\langle REQUEST, r \rangle_{\sigma_c}$ with $r = (o, t, c)$ from a client, an epoch leader first verifies that the request timestamp $t$ is within the client's current watermarks $t_{C_L} < t \leq t_{C_H}$ and maps the request to the respective bucket by hashing the client timestamp and identifier $h_r = H(t||c)$. Each bucket corresponds to a FIFO queue of the received client requests, further referred to as *bucket queue*. We do *not* hash the request payload, as this would allow a malicious client to target a specific bucket by adapting the request payload, mounting load imbalance attacks. If the request falls into the leader's active bucket, the leader also verifies the client's signature $\sigma_c$. A node $i$ discards $r$ if $r$ is already in the corresponding bucket queue.

**PRE-PREPARE.** A leader creates a proposal by adding a batch of requests from its active bucket queues

to a PRE-PREPARE message. The requests that are added in the batch are not immediately deleted but they are marked as pending. This guarantees that the requests maintain their priority if, in the event of an epoch-change, they are not committed. Once leader $i$ gathers enough[3] requests in its current active bucket queues, or if timer $T_{batch}$ expires (since the last batch was proposed by $i$), $i$ adds the non-pending, not preprepared requests from the current active bucket queues in a batch, assigns its next available sequence number $sn$ to the batch (provided $sn$ is within batch watermarks) and sends a PRE-PREPARE message. If $T_{batch}$ time has elapsed and no requests are available, $i$ sends a PRE-PREPARE message with an empty batch. This guarantees progress of the protocol under low load.

A node $j$ accepts a PRE-PREPARE (we say *prepre-pares* the batch and the requests it contains), with sequence number $sn$ for epoch $e$ from node $i$ provided that:

1. the epoch number matches the local epoch number (Alg. 4, Line 82, Alg. 1, Line 5) and $j$ did not preprepare another batch with the same $e$ and $sn$ (Alg. 4, Line 83)

2. node $i$ is in epoch leaders $EL(e)$ (Alg. 4, Line 84, Alg. 4, Line 178)

3. node $i$ leads $sn$ (Alg. 4, Line 84, Alg. 4, Line 179)

4. the batch sequence number $sn$ in the PRE-PREPARE is between a low watermark and high batch watermark: $w < sn \leq W$ (Alg. 4, Line 82, Alg. 1, Line 5)

5. none of the requests in the batch have already been *preprepared* (Alg. 4, Line 85)

6. every request in the batch has timestamp within the current client's watermarks (Alg. 4, Line 86)

7. every request in the batch maps to one of $i$'s active buckets (Alg. 4, Line 87)

8. every request in the batch has a signature which verifies against client's id, i.e., the corresponding public key (Alg. 4, Line 88)

Conditions (1)-(4) are equivalent to checks done in PBFT, whereas conditions (5)-(8) differ from PBFT. Condition (5) is critical for enforcing No Duplication (Property P3, Sec. 2). Conditions (6) (allowing clients to send more than one request concurrently) and (7) (prohibiting

---

[3]determined by the *BatchSize* configuration parameter

malicious leaders to propose requests outside their buckets) are performance related. Condition (8) is the key to Validity (Property P1). As this step may become a CPU bottleneck if performed by all nodes, we use signature sharding as an optimization (see Sec. 5.6).

**Committing a batch.** If node $j$ preprepares the batch, $j$ sends a PREPARE and the protocol proceeds exactly as PBFT (Alg. 4, Lines 93-109). Otherwise, $j$ ignores the batch (which may eventually lead to $j$ entering epoch change). Upon committing a batch, $j$ removes all requests present in the committed batch from $j$'s bucket queues. (Alg. 4, Line 115).

**In-order delivery.** We say that a node $i$ *delivers* a committed batch with sequence number $sn$ once $i$ has delivered all batches with sequence number $sn'$ where $sn' < sn$ (Alg. 4, Lines 119-123). This effectively enforces In-order delivery (Property P6).

Upon delivering a batch with sequence number $sn$, a node outputs DELIVER$(sn_r, r)$ for each request $r$ in the batch. An application running on a replica on top of Mir can now safely execute $r$.

We define a function for assigning request sequence numbers to individual requests. The request sequence number is defined upon delivering the batch which the request is part of. It depends on the relative position of the request within the batch and the total number of requests in all previously delivered batches. Formally, for a batch with sequence number $sn \geq 0$ delivered by some correct node, let $S_{sn}$ be the total number of requests in that batch (possibly 0). Let $r$ be the $k^{th}$ request that a correct node delivers in a batch with sequence number $sn$. Then $sn_r = k$ for $sn = 0$ and $sn_r = k + \sum_{j=0}^{sn-1} S_j$ for $sn > 0$. Notice that when some node $i$ delivers a batch with sequence number $sn$, $S_{sn'}$ is known to $i$ for each $sn'$ where $sn' < sn$, since Mir delivers batches in-order.

## 5.4 Epoch Change

Locally, at node $j$, epoch $e$ can end *graciously*, by exhausting all *maxLen(e)* sequence numbers, or *ungraciously*, if an epoch change timer (corresponding to the PBFT view change timer) at $j$ expires. In the former (gracious) case, a node simply starts epoch $e + 1$ (see also Sec. 5.4.2) when it: (1) locally delivers all sequence numbers in $e$, and (2) reliably delivers the epoch configuration for $e + 1$ (Alg. 4, Line 153). In the latter (ungracious) case, a node first enters an epoch change sub-protocol (Sec. 5.4.1) for epoch $e + 1$ (Alg. 4, Line 125).

It can happen that some correct nodes finish $e$ graciously and some others do not. Such temporary inconsistency may prevent batches from being committed in $e + 1$ even if the primary of $e + 1$ is correct. However, such inconsistent epoch transitions are eventually resolved in subsequent epochs, analogously to PBFT, when some nodes complete the view change sub-protocol and some do not (due to asynchrony). As we show in in Section 7.5, the liveness of Mir is not violated.

### 5.4.1 Epoch Change Sub-protocol

The epoch change sub-protocol is triggered by epoch timeouts due to asynchrony or failures and generalizes PBFT's view change sub-protocol. Upon committing a batch with sequence number $sn$, each correct node starts a timer *ecTimer(sn+1)* for the batch with sequence number $sn + 1$ (Alg. 4, Line 116). The timer is cancelled once the batch is delivered (Alg. 4, Line 122).

If an *ecTimer* for any sequence number expires at node $i$, $i$ enters the epoch-change sub-protocol to move from epoch $e$ to epoch $e + 1$.

In this case, $i$ sends an EPOCH-CHANGE message to the primary of epoch $e + 1$. An EPOCH-CHANGE message follows the structure of a PBFT VIEW-CHANGE message (page 411, [24]) with the difference that it is signed and that there are no VIEW-CHANGE-ACK messages exchanged (to streamline and simplify the implementation similarly to PBFT [22]). The construction of a NEW-EPOCH message (by the primary of $e + 1$) proceeds in the same way as the PBFT construction of a NEW-VIEW message. A node starts epoch $e + 1$ by processing the NEW-EPOCH message the same way a node starts a new view in PBFT by processing a NEW-VIEW message.

However, before entering epoch $e + 1$, each correct node *resurrects* potentially preprepared but uncommitted requests from previous epochs that are not reflected in the NEW-EPOCH message. This is required to prevent losing requests due to an epoch change (due to condition (5) in prepreparing a batch — Sec. 5.3), as not all batches that were created and potentially preprepared before the epoch change were necessarily delivered when starting the new epoch. Resurrecting a request means that each correct node marks the request as not preprepared. The node further marks as not pending any of those requests that where marked pending. (Alg. 4, Line 170) This allows proposing and prepreparing such requests again with a different sequence number. Request resurrection is required for Liveness (P5).

### 5.4.2 Starting a New Epoch

Every epoch $e$, be it gracious or ungracious, starts by the primary reliably broadcasting (using Bracha's classic 3-phase algorithm [18]) the *epoch configuration* information[4] containing: (1) $EL(e)$, the set of epoch leaders for $e$, and (2) identifiers of primary's preferred buckets, which the primary selects based on the oldest request pending at the primary (Alg. 4, Lines 128-141 and Alg. 4, Lines 143-150).

Before starting to participate in epoch $e$ (including processing a potential NEW-EPOCH message for $e$) a node $i$ first waits to reliably deliver the epoch $e$ configuration. In case of gracious epoch change, node $i$ also waits to locally commit all "in-flight" batches pertaining to $e-1$.

## 5.5 Checkpointing (Garbage Collection)

Exactly as in PBFT, Mir uses a checkpoint mechanism to prune the message logs. We consider a sequence number divisible by a predefined configuration parameter as a checkpoint. After a node $i$ commits all batches with sequence numbers up to and including $sn_C$, $i$ sends a $\langle CHECKPOINT, sn_C, H(C')\rangle\sigma_i$ message for the new checkpoint $sn_C$ to all nodes, where $H(C')$ is the hash of the batches with sequence numbers $sn$ in range $sn'_C < sn \leq sn_C$, and $sn'_C$ is the previous checkpoint. Each node collects checkpoint messages until it has $2f+1$ matching ones (including its own), constituting a *checkpoint certificate*, and persists the certificate. At this point, the checkpoint is *stable* and the node can discard the common-case messages from its log for sequence numbers lower than $sn_C$.

Mir advances batch watermarks at checkpoints like PBFT does. Clients' watermarks are also possibly advanced at checkpoints, as the state related to previously delivered requests is discarded. For each client $c$, the low watermark $t_{c_L}$ advances to the highest timestamp $t$ in a request submitted by $c$ that has been delivered, such that all requests with timestamp $t' < t$ have also been delivered. The high watermark advances to $t_{c_H} = t_{c_L} + w_c$, where $w_c$ is the length of the sliding window.

## 5.6 Signature Verification Sharding (SVS)

To offload CPU during failure-free execution (in stable epochs), we implement an optimization where not all nodes verify all client signatures. For each batch, we

---

[4]We optimize the reliable broadcast of an epoch configuration using piggybacking on other protocol messages where applicable.

distinguish $f+1$ *verifier* nodes, defined as the $f+1$ lexicographic (modulo $n$) successors of the leader proposing the batch. Only the verifiers verify client signatures in the batch on reception of a PRE-PREPARE message (condition (8) in Sec. 5.3). Furthermore, we modify the Mir (and thus PBFT) common-case protocol such that a node does not send a COMMIT before having received a PREPARE message from *all* $f+1$ verifiers (in addition to $f$ other nodes and itself). This maintains Validity, as at least one correct node must have verified the client's signature. This way, however, if even a single verifier is faulty, SVS may prevent a batch from being committed. Therefore, we only apply this optimization in stable epochs where all nodes are leaders. In case an (ungracious) epoch change occurs reducing the size of the leaderset, Mir disables SVS. Even though it might seem that SVS gives more opportunity to Byzantine nodes to trigger epoch changes, this is not the case. Since SVS is only enabled when all nodes are leaders, whenever a Byzantine node can trigger an epoch change through SVS, it can also do so by simply not proposing any batches on its own. Such a performance attack could occur in a stable epoch with or without SVS and its impact is examined in Section 9.5.

## 5.7 State Transfer

Nodes can temporarily become unavailable, either due to asynchrony, or due to transient failures. Upon recovery/reconnection, nodes must obtain several pieces of information before being able to actively participate in the protocol again. Mir state transfer is similar to that of PBFT, and here we outline the key aspects of our implementation.

To transfer state, nodes need to obtain current epoch configuration information, the latest stable checkpoint (which occurred at sequence number $h$), as well as information concerning batches having sequence numbers between $h+1$ and the latest sequence number. Nodes also exchange information about committed batches.

The state must, in particular, contain two pieces of information: (1) the current epoch configuration, which is necessary to determine the leaders from which the node should accept proposals, and (2) client timestamps at the latest checkpoint, which are necessary to prevent including already proposed client requests in future batches.

A node $i$ in epoch $e$ initiates state transfer when $i$ receives common-case messages from $f+1$ other nodes with epoch numbers higher than $e$, and $i$ does not transition to $e+1$ for a certain time. Node $i$ obtains this information by broadcasting a $\langle HELLO, ne_i, c_i, b_i\rangle$ mes-

sage, where $ne_i$ is the latest NEW-EPOCH message received by $i$, $c_i$ is the node's last stable checkpoint, and $b_i$ is the last batch $i$ delivered. Upon receipt of a *HELLO* message, another node $j$ replies with its own *HELLO* message, as well as with any missing state from the last stable checkpoint and up to its current sequence number $sn$.

From the latest stable checkpoint, a node can derive the set of $2f + 1$ nodes which signed this stable checkpoint. This also allows a node to transfer missing batches even from one out of these $2f + 1$ nodes, while receiving confirmations of hashes of these batches from $f$ additional nodes (to prevent ingress of batches from a Byzantine node).

We perform further optimizations in order to reduce the amount of data that needs to be exchanged in case of a state transfer. First, upon reconnecting, nodes announce their presence but wait for the next stable checkpoint after state transfer before actively participating in the protocol again. This enables us to avoid transferring the entire state related to requests following the preceding stable checkpoint. Second, the amount of data related to client timestamps that needs to be transmitted can be reduced through only exchanging the root of the Merkle tree containing the client timestamps, with the precise timestamps being fetched only if necessary.

## 5.8 Membership Reconfiguration

While details of membership reconfiguration are outside of the scope of this paper, we briefly describe how Mir deals with adding/removing clients and nodes. Such requests, called *configuration* requests are totally ordered like other requests, but are tagged to be interpreted/executed by nodes. Following the same principle as in Section 5.2, paragraph Buckets, the new configuration should take effect right after the next bucket re-distribution, when no requests are in-flight. This guarantees that all correct nodes are in the same configuration when processing the fist batch after the bucket re-distribution.

## 5.9 Durability (Persisting State)

By default, Mir implementation does not persist state or message logs to stable storage. Hence, a node that crashes might recover in a compromised state — however such a node does not participate in the protocol until the next stable checkpoint which effectively restores the correct state. While we opted for this approach assuming that for few dozens of nodes simultaneous faults of up to

a third of them will be rare, for small number of nodes the probability of such faults grows and with some probability might exceed threshold $f$. Therefore, we optionally persist state pertaining to *sent* messages in Mir, which is sufficient for a node to recover to a correct state after a crash.

We also evaluated the impact of durability with 4 nodes, in a LAN setting, where it is mostly relevant due to small number of nodes and potentially collocated failures, using small transactions. We find that durability has no impact on total throughput, mainly due to the fact that persisted messages are amortized due to batching, Mir parallel architecture and the computation-intensive workload. However, average request latency increases by roughly 300ms.

## 5.10 Implementation Architecture

We implemented Mir in Go. Our implementation is multi-threaded and inspired by the *consensus-oriented parallelism* (COP) architecture previously applied to PBFT to maximize its throughput on multicore machines [16]. Specifically, in our implementation, a separate thread is dedicated to managing each batch during the common case operation, which simplifies Mir code structure and helps maximize performance. We further parallelize computation-intensive tasks whenever possible (e.g., signature verifications, hash computations). The only communication in common case between Mir threads pertains to request duplication prevention (rule (6) in accepting PRE-PREPARE in Sec. 5.3) — the shared data structures for duplication prevention are hash tables, synchronized with per-bucket locks; instances that handle requests corresponding to different leaders do not access the same buckets. The only exception to the multi-threaded operation of Mir is during an ungracious epoch-change, where a designated thread (Mir Manager) is responsible for stopping worker common-case threads and taking the protocol from one epoch to the next. This manager thread is also responsible for sequential batch delivery and for checkpointing, which, however, does not block the common-case threads processing batches.

Our implementation also parallelizes network access using a configurable number of independent network connections between each pair of nodes. This proves to be critical in boosting Mir performance beyond seeming bandwidth limitations in a WAN that stem from using a single TCP/TLS connection.

In addition to multiple inter-node connections, we use an independent connection for handling client requests. As a result, the receipt of requests is independent of

the rest of the protocol — we can safely continue to receive client requests even if the protocol is undergoing an epoch change. Our implementation can hence seamlessly use, where possible, separate NICs for client's requests and inter-node communication to address DoS attacks [25].

# 6 Pseudocode

In this section we introduce Mir pseudocode. We first present PBFT [24] pseudocode to demonstrate the common message flow in the common case of the two protocols. Experienced readers familiar with PBFT are encouraged to fast forward to Mir (Algorithm 4).

Each node executes its own instance of the algorithm described by the pseudocode. The node atomically executes each **upon** block exactly once for each assignment of values satisfying the block's triggering condition.

For better readability we do not include batching in the pseudocode. Implementing batching is trivial by replacing requests with batches of requests, except request handling (Algorithm 4, lines 55- 62). Moreover, whenever appropriate, instead of performing a request-specific action on a batch, we perform this action on all requests in a batch, like request validity checks in PRE-PREPARE (Algorithm 4, lines 86- 88) and request resurrection (Algorithm 4, lines 162-176). In the context of request-specific validity checks, we consider the whole batch invalid if any of the contained requests fails its validity check. Finally, *PreprepareTimeout* corresponds to $T_{batch}$ and, with batching enabled, condition in Algorithm 4, line 66 should be replaced with checking either if $T_{batch}$ has ellapsed or if there exist enough requests for a batch.

# 7 Mir Correctness

In this section we outline the Mir correctness proof, proving TOB properties as defined in Section 2. We pay particular attention to Liveness (Section 7.5), as we believe it is the least obvious out of four Mir TOB properties to a reader knowledgeable in PBFT. Where relevant, we also consider the impact of the signature verification sharding (SVS) optimization (Sec. 5.6).

## 7.1 Validity (P1)

(P1) **Validity:** If a correct node delivers $r$, then some client broadcast $r$.

*Proof (no SVS).* We first show that Validity holds, without signature verification sharding. If a correct node delivers $r$, then at least $n - f$ nodes sent COMMIT for a batch which contains $r$ which includes at least $n - 2f \geq f + 1$ correct nodes (Sec. 3). Similarly, if a correct node sends COMMIT for a batch which contains $r$, then at least $n - 2f \geq f + 1$ correct nodes sent PREPARE after prepreparing a batch which contains $r$ (Sec. 5.3). This implies at least $f + 1$ correct nodes executed Condition (8) in Sec. 5.3 and verified client's signature on $r$ as correct. Validity follows. □

*Proof (with SVS).* With signature verification sharding (Sec. 5.6), clients' signatures are verified by at least $f + 1$ *verifier* nodes belonging to the leaderset, out of which at least one is correct. As no correct node sends COMMIT before receiving PREPARE from all $f + 1$ *verifier* nodes (Sec. 5.6), no request which was incorrectly signed by a client can be committed and, subsequently, delivered. Validity follows. □

## 7.2 Agreement (Total Order) (P2)

(P2) **Agreement:** If two correct nodes deliver requests $r$ and $r'$ with sequence number $sn$, then $r = r'$.

*Proof.* Assume by contradiction that there are two correct nodes $i$ and $j$ which deliver, respectively, $r$ and $r'$ with the same sequence number $sn$, such that $r \neq r'$. Without loss of generality, assume $i$ delivers $r$ with $sn$ before $j$ delivers $r'$ with $sn$ (according to a global clock not accessible to nodes), and let $i$ (resp., $j$) be the first correct node that delivers $r$ (resp., $r'$) with $sn$.

By the way we compute request sequence numbers (see Sec. 5.3, In-order delivery), the fact that $i$ and $j$ deliver different requests at the same (request) sequence number implies they commit different batches with same (batch) sequence number. Denote these different batches by $B$ and $B'$, respectively, and the batch sequence number by $bsn$.

We distinguish several cases depending on the mechanism by which $i$ (resp., $j$) commits $B$ (resp $B'$). Namely, in Mir, $i$ can commit *req* contained in batch $B$ in one of the following ways (commit possibilities (CP)):

CP1  by receiving a quorum ($n - f$) of matching COMMIT messages in the common case of an epoch for a fresh batch $B$ (a fresh batch here is a batch for which a leader sends a PRE-PREPARE message — see Sec. 3 and Sec. 5.3)

## Algorithm 1 Common

```
 1: function IsPrimary(id, view, num_nodes) :
 2:     return id = view  mod num_nodes;
 3:
 4: function Valid(local_view, view, seq_no, low, high) :
 5:     if (local_view = view)  and (low <= seq_no < high) then
 6:         return True;
 7:     else
 8:         return False;
 9:     end if
10:
11: function GetOldest(S_1, S_2) :
12:     Returns the oldest entry in set S_1 \ S_2.
13:
```

## Algorithm 2 PBFT [24]

```
 1: import Common
 2: import PbftViewChange
 3:
 4: Parameters:
 5:     id                                                              // The node identity
 6:     f                                                              // Number of faults tolerated
 7:     RequestTimeout          // Timeout to prevent waiting indefinitely for q request to commit
 8:     w                                                    // Low watermark, advances at checkpoints
 9:     W                                                   // High watermark, advances at checkpoints
10:
11: Struct Request contains
12:     bytes o                                                         // Request payload
13:     int t                                                          // Client timestamp
14:     bytes c                                                       // Client public key (ID)
15:
16: Init:
17:     lv ← 0                                                        // Local view number
18:     next ← 0                                              // The next available sequence number
19:     R ← ∅                                                        // The set of received requests
20:     Preprepare_msgs ← {}      // A map from (view, sequence number) pairs to PRE-PREPARE messages, initially ⊥
21:     Prepare_msgs ← {}           // A map from (view, sequence number) pairs to a set of unique PREPARE messages
22:     Commit_msgs ← {}            // A map from (view, sequence number) pairs to a set of unique COMMIT messages
23:     RequestTimeouts ← {}                                          // A map from requests to timers
24:
25: // Handling client request
26: upon receiving ⟨REQUEST, r⟩_{σ_c}
27:     such that SigVer(r, σ_c, c)
28:     and  not (r' in R s.t. r'.c = r.c  and r'.t ≠ r.t) do
29:     R ← R ∪ {r}
30:     RequestTimeouts[r] ← schedule RequestTimeout
31:
```

**Algorithm 2** PBFT (continues)

32: // Sending new PRE-PREPARE message
33: **upon** $|R| > 0$ **and** $w <= next < W$
34:  **and** common.IsPrimary(id, lv, N) **do**
35:     $r \leftarrow common.GetOldest(R, \emptyset)$
36:     Send $\langle PRE\text{-}PREPARE, lv, next, r, id \rangle$ to all nodes
37:     $next \leftarrow next + 1$
38:
39: // Handling PRE-PREPARE message and sending PREPARE message
40: **upon receiving** $pp \leftarrow \langle PRE\text{-}PREPARE, v, n, r, i \rangle$
41:     **such that** $common.Valid(lv, v, n, w, W)$
42:     **and** $common.IsPrimary(i, v, N)$
43:     **and** $Preprepare\_msgs[v, n] = \perp$
44:     **and** $r$ **in** $R$ **do**
45:     $Preprepare\_msgs[v, n] \leftarrow pp$
46:     send $\langle PREPARE, v, n, D(r), id \rangle$ to all nodes
47:
48: // Handling PREPARE message
49: **upon receiving** $p \leftarrow \langle PREPARE, v, n, D(r), i \rangle$
50:     **such that** $D(Preprepare\_msgs[v, n].r) = D(r)$
51:     **and** $common.Valid(lv, v, n, w, W)$ **do**
52:     $Prepare\_msgs[v, n] \leftarrow Prepare\_msgs[v, n] \cup \{p\}$
53:
54: // Sending COMMIT message
55: **upon** $|Prepare\_msgs[lv, n]| = 2f + 1$ **do**
56:     $r \leftarrow Preprepare\_msgs[lv, n].r$
57:     send $\langle COMMIT, lv, n, D(r), id \rangle$ to all nodes
58:
59: // Handling COMMIT message
60: **upon receiving** $c \leftarrow COMMIT, v, n, D(r), i \rangle$
61:     **such that** $D(Preprepare\_msgs[v, n].r) = D(r)$
62:     **and** $common.Valid(lv, v, n, w, W)$ **do**
63:     $Commit\_msgs[v, n] \leftarrow Commit\_msgs[v, n] \cup \{c\}$
64:
65: // Delivering request
66: **upon** $|Commit\_msgs[lv, n]| = 2f + 1$ **do**
67:     $r \leftarrow Preprepare\_msgs[v, n].r$
68:     $R \leftarrow R \setminus \{r\}$
69:     $Deliver(n, r)$
70:     **cancel** $RequestTimeouts[r]$
71:
72: // View change on request timeout
73: **upon** $RequestTimeout$ **do**
74:     $lv \leftarrow lv + 1$
75:     $PbftViewChange.ViewChange()$
76:

**Algorithm 3** PBFT ViewChange

```
 1: import Common
 2:
 3: Parameters:
 4:     N                                                                    // Number of nodes
 5:     f                                                                    // Number of faults
 6:     id                                                                   // The node identity
 7:     lv                                                                   // Local view number
 8:     P               // Map form sequence number to Entry struct for the latest prepared request in previous views
 9:     Q          // Map form sequence number to all Entry structs for a unique preprepared request in previous views
10:     C                                                                    // Local checkpoints
11:     h                                                                    // Latest stable checkpoint
12:
13: Init:
14:     Sset ← {}                                              // A map from node id to ViewChange message
15:     Xset ← {}                                            // A map from sequence number to selected value
16:     cp ← ⊥                                        // Highest stable checkpoint available by a f+1 nodes
17:
18: Struct Request contains
19:     n                                                                    // Sequence number
20:     d                                                                    // Request digest
21:     v                                                                    // View
22:
23: // Handling VIEWCHANGE message
24: upon receiving  m ← VIEWCHANGE, v, h, C, P, Q, i, σ_i⟩
25:        such that SigVer(m, σ_i, i.pk)
26:     V[i] ← m
27:     if |Sset| ≥ 2f + 1
28:         CalculateHighCheckpoint(Sset)
29:         CalculateXset(Sset) ^5
30:         if Xset ≠ {}                                              // If the Xset is successfully calculated
31:             send ⟨NEWVIEW, v, Sset, Xset, cp, id, σ_{id}⟩ to all nodes
32:         end if
33:     end if
34:
35: // Handling NEWVIEW message
36: upon receiving  m ← NEWVIEW, v, S, X, cp', i, σ_i⟩
37:        such that SigVer(m, σ_i, i.pk)
38:     CalculateHighCheckpoint(S)
39:     CalculateXset(S)
40:     if Xset = X and cp = cp'                                           // Verify NEWVIEW
41:         for all (n, r) ∈ Xset do
42:             send ⟨PREPARE, v, n, D(r), id⟩ to all nodes
43:         end for
44:     end if
45:
46: function ViewChange() :
47:     lv ← lv + 1                                                        // Advance local view
48:     p ← lv mod N;                                                       // Find the new primary
49:     send ⟨VIEWCHANGE, lv, h, C, P, Q, id, σ_{id}⟩ to p
50:
51: function CalculateHighCheckpoint(V) :
52:     cp ← cp'|cp' the highest checkpoint in m.C(∀m ∈ Sset) and at least f + 1 nodes have a checkpoint in cp'.
53:
```

---

**Algorithm 3** PBFT ViewChange (continues)

---

54: **function** *CalculateXset(V)* :
55:     $L \leftarrow$ the highest sequence number in $m.P (\forall m \in Sset)$
56:     **for all** $n$  **such that** $cp < n \leq L$ **do**
57:         **if** $\exists m \in Sset$ with $\langle n, d, v \rangle \in m.P$
58:             **such that** $\exists 2f + 1$ messages $m' \in Sset$
59:                 **such that** $m'.h < n$
60:                 **and** $\forall \langle n, d', v' \rangle \in m'.P$
61:                     **such that** $v' < v$ **or** $(v' = v$ **and** $d' = d)$
62:             **and** $\exists f + 1$ messages $m' \in Sset$
63:                 **such that** $\exists \langle n, d', v' \rangle \in m'.Q$
64:                     **such that** $v' \geq v$ **and** $d' = d$
65:             $X[n] \leftarrow$ request with digest $d$               // Request with digest $d$ could have been prepared for $n$
66:         **else if** $\exists 2f + 1$ messages $m \in Sset$
67:             **such that** $m.h < n$ **and** $m.P$ has no entry for $n$
68:             $X[n] \leftarrow \perp$                                // No request could have been prepared for $n$
69:         **else**
70:             $Xset \leftarrow \{\}$                                 // Not enough VIEWCHANGE messages
71:             **return**
72:         **end if**
73:     **end for**
74:

---

CP2 by receiving a quorum $(n - f)$ of matching COM-MIT messages following an ungracious epoch change, where NEW-EPOCH message contains $B$ (Sec. 5.4.1),

CP3 via the state transfer sub-protocol (Sec. 5.7).

As $i$ is the first correct node to commit request $r$ with $sn$ (and therefore batch $B$ with $bsn$), it is straightforward to see that $i$ cannot commit $B$ via state transfer (CP3). Hence, $i$ commits $B$ by CP1 or CP2.

We now distinguish several cases depending on the CP by which $j$ commits $B'$. In case $j$ commits $B'$ by CP1 or CP2, since Mir common case follows the PBFT common case, and Mir ungracious epoch change follows PBFT view change — a violation of Agreement in Mir implies a violation of Total Order in PBFT. A contradiction.

The last possibility is that $j$ commits $B'$ by CP3 (state transfer). Since $j$ is the first correct node to commit $B'$ with $bsn$, $j$ commits $B'$ after a state transfer from a Byzantine node. However, since (1) Mir CHECKPOINT messages (see Sec. 5.5) which are the basis for stable checkpoints and state transfer (Sec. 5.7) are signed, and (2) stable checkpoints contain signatures of $2f + 1$ nodes including at least $f + 1$ correct nodes, $j$ is not the first correct node to commit $B'$ with $bsn$. A contradiction. $\square$

## 7.3 No Duplication (P3)

**(P3) No duplication:** If a correct node delivers request $r$ with sequence numbers $sn$ and $sn'$, then $sn = sn'$.

*Proof.* No-duplication stems from the way Mir prevents duplicate prepreparares (condition (5) in accepting PRE-PREPARE, as detailed in Sec. 5.3).

Assume by contradiction that two identical requests $req$ and $req'$ exist such that $req = req'$ and correct node $j$ delivers $req$ (resp., $req'$) with sequence number $sn$ (resp., $sn'$) such that $sn \neq sn'$.

Then, we distinguish the following exhaustive cases:

- *(i) req* and *req'* are both delivered in the same batch.

- *(ii) req* and *req'* are delivered in different batches.

In case *(i)*, assume without loss of generality that $req$ precedes $req'$ in the same batch. Then, by condition (5) for validating a PRE-PREPARE (Sec. 5.3), no correct node prepreparares $req'$ and all correct nodes discard the batch which hence cannot be delivered, a contradiction.

In case *(ii)* denote the batch which contains $req$ by $B$ and the batch which contains $req'$ by $B'$. Denote the set of at least $n - f \geq 2f + 1$ nodes that prepare batch $B$ by $S$ and the set of at least $n - f \geq 2f + 1$ that prepare batch $B'$ by $S'$. Sets $S$ and $S'$ intersect in at least $n - 2f \geq f + 1$ nodes out of which at least one is correct,

**Algorithm 4** Mir

```
 1: import Common
 2: import PbftViewChange
 3: import ReliableBroadcast
 4:
 5: Parameters:
 6:     id                                                          // The node identity
 7:     f                                                           // Number of faults tolerated
 8:     w                                                           // Low watermark, advances at checkpoints
 9:     W                                                           // High watermark, advances at checkpoints
10:     NumBuckets                                                  // Number of buckets
11:     BucketsPerLeader                    // The number of buckets per leader when all nodes are leaders
12:     RedistributionPeriod                                        // Bucket re-distribution period
13:     EphemeralEpLen                              // Number of sequence numbers in an ephemeral epoch
14:
15: Struct Request contains
16:     o                                                           // Request payload
17:     t                                                           // Client timestamp
18:     c                                                           // Client identity (public key)
19:
20: Struct Client contains
21:     H                                            // Client high watermark, advances at checkpoint
22:     L                                            // Client low watermark, advances at checkpoint
23:
24: Struct EpochConfig contains
25:     First                                             // First sequence number of the epoch
26:     Last                                              // Last sequence number of the epoch
27:     Leaders                                           // List of leaders of the epoch
28:     PrimaryBuckets                                    // Buckets the primary chose for itself
29:
30: Init:
31:     le ← 0                                                      // Local epoch number
32:     next ← id                                                  // The next available sequence number
33:     Buckets ← Set of NumBuckets empty buckets      // Each bucket is a FIFO queue of received requests
34:     Clients ← {}                       // A map from client identity (public key) to a Client structure
35:     Preprepare_msgs ← {}          // A map from (epoch, sequence number) pairs to PRE-PREPARE messages
36:     Prepare_msgs ← {}         // A map from (epoch, sequence number) pairs to a set of unique PREPARE messages
37:     Commit_msgs ← {}          // A map from (epoch, sequence number) pairs to a set of unique COMMIT messages
38:     Pendng ← ∅                                     // A set of proposed but not committed requests
39:     Preprepared ← ∅                               // A set of preprepared requests to prevent duplicates
40:     committed ← {}            // A map from (epoch, sequence number) pairs to committed requests, initially ⊥
41:     delivered ← {}                       // A map from (epoch, sequence number) booleans
42:     EpochChangeTimeouts                  // List of timers per sequence number for epoch change
43:     EpochConfig ← []                                           // List of epoch configurations
44:     for all bucket ∈ Buckets do
45:         bucket ← ∅
46:     end for
47:     EpochConfig[0].First = 0
48:     EpochConfig[0].Last = ∞
49:     EpochConfig[0].Leaders = Nodes
50:     EpochConfig[0].PrimaryBuckets = arbitrary ⌈NumBuckets/Nodes⌉ buckets
51:     ActiveBucketAssignment(0, EpochConfig[0])
52:     EpochChangeTimeouts[0] ← start EpochChangeTimeout    // Start a timer for the first sequence number
53:     start PreprepareTimeout                              // Start a timer for a new preprepare
54:
```

**Algorithm 4** Mir (continues)

55: // Handling client request
56: **upon receiving** $\langle REQUEST, r \rangle_{\sigma_c}$
57:     **such that** $SigVer(r, \sigma_c, r.c)$
58:     **and** $Clients[r.c].L <= r.t < Clients[r.c].H$ **do**
59:         $bucket \leftarrow GetBucket(H(t||c))$
60:     **if** $\nexists r' \in bucket : r'.c = r.c \wedge r'.t = r.t$ **then**
61:             $bucket.append(r)$
62:     **end if**
63:
64: // Sending new PRE-PREPARE message
65: **upon** $(|ActiveBuckets(i, le, next) \setminus Pending| > 0$ **or** PrepareTimeout)          // The active bucket is not empty
66:                                                                                              // or the timeout for a new PRE-PREPARE has elapsed
67:     **and** $w <= next < W$                                                                   // The next seq no is in the watermark window
68:     **and** $ActiveDistribution(le, next)$                                                    // All seq nos from the previous re-distribution are delivered
69:     **and** $n \le EpochConfig[le].Last$ **do**                                               // The next seq no is in the current epoch
70:     **if** $|ActiveBuckets(i, le, next) \setminus Pending| > 0$ **then**                      // The active bucket is not empty
71:         $r \leftarrow common.GetOldest(ActiveBuckets(i, le, next) \setminus Pendng, Preprepared)$
72:         $Pending \leftarrow Pending \cup \{r\}$
73:     **else**
74:         $r \leftarrow \bot$                                                                   // Sending PRE-PREPARE with special nil request
75:     **end if**
76:     send $\langle PRE\text{-}PREPARE, le, next, r, id \rangle$ to all nodes
77:     $next \leftarrow next + |EpochConfig[le].Leaders)|$
78:     **reset** PrepareTimeout
79:
80: // Handling PRE-PREPARE message and sending PREPARE message
81: **upon receiving** $pp \leftarrow \langle PRE\text{-}PREPARE, e, n, r, i \rangle$
82:     **such that** $common.Valid(le, e, n, w, W)$                                              // Valid sequence number and epoch number Sec. 5.3(1),(4)
83:     **and** $Preprepare\_msgs[e, n] = \bot)$                                                   // No other batch is preprepared with *sn* in epoch *e* Sec. 5.3(1)
84:     **and** $Leads(i, e, n)$                                                                   // Node *i* is in the leadset of epoch *e* Sec. 5.3(2) and leads *sn* Sec. 5.3(3)
85:     **and** $H(r.o||r.t||r.c)$ **not in** $Preprepared$                                        // The request is not already preprepared Sec. 5.3(5)
86:     **and** $Clients[r.c].L <= r.t < Clients[r.c].H$        // The client timestamp is within the client's watermark window Sec. 5.3(6)
87:     **and** $H(r.t||r.c)$ **in** $ActiveBuckets(i, e, n)$                                      // The request belongs to an active bucket for *i* Sec. 5.3(7)
88:     **and** $SigVer((r, r.\sigma_c, r.c)$ **do**                                               // The request was a valid signature Sec. 5.3(8)
89:     $Preprepared \leftarrow Preprepared \cup \{r\}$
90:     $Preprepare\_msgs[e, n] \leftarrow pp$
91:     send $\langle PREPARE, v, n, D(r), id \rangle$ to all nodes
92:
93: // Handling PREPARE message
94: **upon receiving** $p \leftarrow \langle PREPARE, e, n, D(r), i \rangle$
95:     **such that** $D(Preprepare\_msgs[e, n].r) = D(r)$
96:     **and** $common.Valid(le, e, n, w, W)$ **do**
97:     $Prepare\_msgs[e, n] \leftarrow Prepare\_msgs[e, n] \cup \{p\}$
98:
99: // Sending COMMIT message
100: **upon** $|Prepare\_msgs[le, n]| = 2f + 1$ **do**
101:     $r \leftarrow Preprepare\_msgs[e, n].r$
102:     send $\langle COMMIT, le, n, D(r), id \rangle$ to all nodes
103:
104: // Handling COMMIT message
105: **upon receiving** $c \leftarrow \langle COMMIT, e, n, D(r), i \rangle$
106:     **such that** $D(Preprepare\_msgs[e, n].r) = D(r)$
107:     **and** $common.Valid(le, e, n, w, W)$ **do**
108:     $Commit\_msgs[e, n] \leftarrow Commit\_msgs[e, n] \cup \{c\}$
109:

**Algorithm 4** Mir (continues)

```
110: // Committing a request
111: upon |Commit_msgs[e,n]| = 2f + 1 do
112:     r ← Preprepare_msgs[e,n].r
113:     committed[e,n] ← r
114:     Pending ← Pending \ {r}
115:     GetBucket(H(r.t||r.c)).remove(r)                                    // Removing request from bucket
116:     EpochChangeTimeouts[n + 1] ← start EpochChangeTimeout    // Epoch-change timer for the next sequence number
117:
118: // In-order request delivery
119: upon committed[le,n] ≠ ⊥ and delivered[n − 1] do        // The previous sequence number must be already delivered
120:     Deliver(n, r)
121:     delivered[n] ← True
122:     cancel EpochChangeTimeouts[n]                                    // Cancelling the new epoch-change timer for n
123:
124: // Epoch change timeout
125: upon EpochChangeTimeout do
126:     PBFTViewChange.ViewChange()                                     // Algorithm 3: PBFT view change
127:
128: // Reliable broadcasts of new epoch configuration on gracious epoch change
129: upon delivered[EpochConfig[e].Last]                    // all sequence numbers of the epoch are delivered
130:   and common.IsPrimary(id, e+1, N) do
131:     EpochConfig[e + 1].Leaders ← EpochConfig[e].Leaders ∪ {id}     // the primary adds itself in the leaderset
132:     EpochConfig[e + 1].PrimaryBuckets
133:         ← ⌈NumBuckets/Nodes⌉ buckets containing the oldest requests       // primary's preferred buckets
134:     EpochConfig[e + 1].First ← EpochConfig[e].Last + 1
135:     ifEpochConfig[e + 1].Leaders = Nodes then                      // if all nodes are in the leaderset
136:         EpochConfig[e + 1].Last ← ∞                               // the next epoch is stable
137:     else
138:         EpochConfig[e + 1].Last
139:             ← EpochConfig[e + 1].First + ephemeralEpLen
140:     end if
141:     ReliableBroadcast.Broadcast(EpochConfig[e + 1], e + 1)
142:
143: // Reliable broadcast of new epoch configuration on ungracious epoch change
144: upon sending PBFT NEW-EPOCH message for epoch e + 1 do
145:     EpochConfig[e + 1].Leaders ← ShrinkingLeaderset(e + 1, id)
146:     EpochConfig[e + 1].PrimaryBuckets
147:         ← ⌈NumBuckets/Nodes⌉ buckets containing the oldest requests
148:     EpochConfig[e + 1].First ← EpochConfig[e].Last + 1
149:     EpochConfig[e + 1].Last ← EpochConfig[e + 1].First + ephemeralEpLen
150:     ReliableBroadcast.Broadcast(EpochConfig[e + 1], e + 1)
151:
152: // Gracious epoch change
153: upon ReliableBroadcast.Delivered(EpochConfig, e) and le = e do
154:     EpochConfig[e] ← EpochConfig
155:     if ∃k : EpochConfig[e].Leaders[k] = id then
156:         next ← EpochConfig[e].First + k
157:     end if
158:     ActiveBucketAssignment(e, EpochConfig)
159:
```

**Algorithm 4** Mir (continues)

160: **upon** sending or receiving PBFT NEW-EPOCH message **do**
161:     Process the message according to Algorithm 3
162:     // Request resurrection
163:     **for all** $r \in Preprepared$ **do**
164:         **if** r not in NEW-EPOCH **then**
165:             $Preprepared \leftarrow Preprepared \setminus \{r\}$         // Uncommitted requests, inserted in the set in a previous epoch
166:         **end if**
167:     **end for**
168:     **for all** $r \in Preprepared$ **do**
169:         **if** r not in NEW-EPOCH **then**
170:             $Pending \leftarrow Pending \setminus \{r\}$         // Uncommitted requests, inserted in the set in a previous epoch
171:         **end if**
172:     **end for**
173:     **for all** $r \in$ PBFT NEW-EPOCH **do**
174:         $Preprepared \leftarrow Preprepared \cup \{r\}$         // Marking again the requests in the message as preprepared
175:     **end for**
176:
177: **function** $Leads(i,e,n)$ **:**         // Returns $True$ if $i$ is leader of $n$ in epoch $e$
178:     **if** $i$ in $EpochConfig[e].Leaders$ **then**
179:         **return** $((EpochConfig[e].First + n) \mod |EpochConfig[e].Leaders|) = i$
180:     **else**
181:         **return** False
182:     **end if**
183:
184: **function** $GetBucket(hash)$ **:**
185:     Returns the bucket containing requests $r$ such that $H(r.t||r.c) = hash$.
186:
187: **function** $ActiveBucketAssignment(e, EpochConfig)$ **:**
188:     Evenly partition $Buckets \setminus EpochConfig.PrimaryBuckets$
189:     among $EpochConfig.Leaders \setminus \{i : common.IsPrimary(i,e,N)\}$
190:
191: **function** $ActiveBuckets(i,e,n)$ **:**
192:     Returns the union of buckets which are active for node $i$ in epoch $e$ and sequence number $n$
193:
194: // ActiveDistribution returns true if all the sequence numbers from the previous re-distribution are delivered
195: **function** $ActiveDistribution(e,n)$ **:**
196:     $period \leftarrow RedistributionPeriod$
197:     $distribution \leftarrow \lceil n - (EpochConfig[e-1].Last)/period \rceil$
198:     **return** $delivered[EpochConfig[e-1].Last + (distribution - 1) * period]$
199:
200: **function** $ShrinkingLeaderset(e,i)$ **:**
201:     $e_{last} \leftarrow$ the last epoch for which $i$ has the configuration
202:     $Leaders \leftarrow EpochConfig[e_{last}].Leaders \cup \{i\}$
203:     $RemovedLeaders \leftarrow$ a random set of $min((e'-e),1)$ nodes from $EpochConfig[e_{last}].Leaders \setminus \{i\}$
204:     **return** $Leaders \setminus RemovedLeaders$
205:

say node $i$. Assume without loss of generality that $i$ preprepares $B$ before $B'$. Then, the following argument holds irrespectively of whether $i$ delivers batch $B$ before $B'$, or vice versa: as access to data structure responsible for implementing condition (5) is synchronized with per-bucket locks (Sec. 5.10) and since $req$ and $req'$ both map to the same bucket, as their hashes are identical, $i$ cannot preprepare $req'$ and hence cannot prepare batch $B'$ which cannot be delivered — a contradiction. $\square$

It is easy to see that signature verification sharding optimization does not impact the No-Duplication property.

## 7.4 Totality (P4)

**Lemma 1.** *If a correct node delivers a sequence number sn, then every correct node eventually delivers sn.*

*Proof.* Assume, by contradiction, that a correct node $j$ never delivers any request with $sn$. We distinguish 2 cases:

1. $sn$ becomes part of a stable checkpoint of a correct node $k$. In this case, at the latest after GST $j$ enters the state transfer protocol (see Sec. 5.7), and transfers the missing batches, including the batch with sequence number $sn$, from some correct node. Such a node exists, because at least $k$ is such a node. Moreover, $j$ gets batch hash confirmations from $f$ additional nodes that signed the stable checkpoint $sn$ belongs to. At this point $j$ can deliver all sequence numbers up to the stable checkpoint, obtained with the state transfer protocol, including $sn$, because $j$ transfers all sequence numbers up to the stable checkpoint without gaps. A contradiction.

2. $sn$ never becomes part of a stable checkpoint. Then, the start of the watermark window will never advance past $sn$, and all correct nodes, at latest when exhausting the current watermark window, will start infinitely many ungracious epoch changes without any of them committing, and therefore delivering, any requests. Correct nodes will always eventually exhaust the sequence numbers in their current watermark window, since even in the absence of new client requests, correct leaders periodically propose a special nil request (in practice, an empty batch) (see Algorithm 4, line 74). Infinitely many ungracious epoch changes without committing any requests, however, is a contradiction to PBFT liveness. $\square$

(P4) **Totality:** If a correct node delivers request $r$, then every correct node eventually delivers $r$.

*Proof.* Let $i$ be a correct node that delivers $r$ with sequence number $sn$. Then, by (P2) Agreement, no correct node can deliver another $r' \neq r$ with sequence number $sn$. Therefore, all other correct nodes will either deliver $r$ with $sn$ or never deliver $sn$. The latter is a contradiction to Lemma 1, since $i$ delivered some request with $sn$, all correct nodes deliver some request with $sn$. Totality follows. $\square$

## 7.5 Liveness (P5)

We first prove a number of auxiliary Lemmas and then prove liveness.

**Lemma 2.** *In an execution with a finite number of epochs, the last epoch $e_{last}$ is a stable epoch.*

*Proof.* Assume by contradiction that $e_{last}$ is not stable, this implies either:

1. a gracious epoch change from $e_{last}$ at some correct node and hence, $e_{last}$ is not the last – a contradiction; or

2. ungracious epoch change from $e_{last}$ never completes — since Mir ungracious epoch change protocol follows PBFT view change protocol, this implies liveness violation in PBFT. A contradiction.

$\square$

**Lemma 3.** *If a correct client broadcasts request $r$, then every correct node eventually receives $r$ and puts it in the respective bucket queue or delivers $r$.*

*Proof.* The lemma follows by assumption of a synchronous system after GST and by the correct client sending and periodically re-sending request to all nodes until a request is delivered (see Section 5.1). $\square$

**Lemma 4.** *If, after GST, all correct nodes start executing the common-case protocol in a non-stable epoch $e$ before time $t$, then there exists a $\Delta$, such that if a correct leader proposes a batch $B$ before time $t$ and no correct node enters an epoch change before $t + \Delta$, every correct node commits $B$.*

*Proof.* Let $\delta$ be the upper bound on the message delay after GST and let a correct leader propose a request $r$ before $t$. By the common-case algorithm without SVS (there is

no SVS in a non-stable epoch $e$) all correct nodes receive at least $2f + 1$ COMMIT messages for $r$ before $t + 3\delta$ (time needed to transmit PRE-PREPARE, PREPARE and COMMIT). All correct nodes will accept these messages, since they all enter epoch $e$ by time $t$. As every correct node receives at least $2f + 1$ COMMITs, every correct node commits $r$ by $t + 3\delta$. Therefore, $\Delta = 3\delta$. $\square$

**Lemma 5.** *If a correct node commits a batch which includes request $r$, then $i$ eventually delivers $r$.*

*Proof.* Let us assume that $r$ is committed by $i$ in a batch with sequence number $bsn$. Then, for $bsn$ not to be delivered, there exists some batch with sequence number $bsn' < bsn$ which is never delivered by $i$. Otherwise, $i$ commits all batches with a sequence number $bsn' < bsn$ and, therefore, $i$ can deliver the batch with sequence number $bsn$ which includes $r$ (see Alg. 4, Line 119).

We distinguish two cases.

1. There exists some correct node that delivers a batch with sequence number $bsn'$.

2. No correct node delivers a batch with sequence number $bsn'$.

In the first case, by Lemma 1, every correct node, including $i$, delivers (and thus commits) a batch with sequence number $bsn'$. To deliver a batch, a node must have committed it (see Alg. 4, Line 119).

In the second case, similarly to the argumentation in the second case in Lemma 1, the start of the watermark window in any correct node will never advance beyond $bsn'$, leading to infinitely many epoch changes, without committing any request. A contradiction to PBFT liveness. $\square$

**Lemma 6.** *During a single epoch, a correct node does not propose the same request more than once.*

*Proof.* After a request $r$ is proposed the first time in an epoch, until the end of that epoch there are two mutually exclusive cases. A proposed request $r$ is either pending or committed (included in a committed batch) (Alg. 4: Line 72, Alg. 4: Line 114).

In the first case, a correct node does not propose $r$ because it is marked as pending (Alg. 4: Line 71). $r$ can be un-marked pending without being committed only with request resurrection. However, resurrection can only occur during an epoch change and thus the second proposal cannot happen in the same epoch as the first.

In the second case, $r$, upon being committed in a batch at some correct node $i$, $i$ removes $r$ from its bucket queue

(Alg. 4: Line 115). If $r$ is committed by $i$ then $r$ is also preprepared by $i$ and $i$ will not propose $r$ again (Alg. 4: Line 71). Moreover, $i$ does not remove $r$ from the preprepared set with request resurrection within an epoch, because, as in the previous case, this requires an epoch change.

In either of the cases $i$ does not propose $r$ again in the same epoch. $\square$

**Lemma 7.** *If a correct node proposes a batch $b$ in epoch $e$ with sequence number $sn$, then no correct node delivers a batch $b' \neq b$ with $sn$ in epoch $e$.*

*Proof.* Let us denote by $i$ the node that proposes $b$ in epoch $e$.

Assume, by contradiction, that some correct node $j$, including the case where $i = j$, delivers $b'$ with $sn$ in epoch $e$. Then $j$ must have preprepared $b'$ in the same epoch $e$; otherwise $j$ cannot commit $b'$ in $e$ and, consequently, delivered it. Since $i$ is the leader of sequence number $sn$ in epoch $e$, $j$ does not preprepare a batch proposed by any node other than $i$ in $e$ (see Alg. 4, Line 84). Therefore, $j$ can preprepare $b'$ with sequence number $sn$ only if $i$ proposes $sn$. However, $i$, being a correct node, proposes only one batch per sequence number, and by lemma statement we know that that is $b$. This is a contradiction to $b' \neq b$. $\square$

**Lemma 8.** *When node $i$ is assigned the bucket of request $r$ in epoch $e$, $i$ has not proposed $r$ in $e$ and $i$ has marked $r$ as preprepared, then $i$ has delivered $r$.*

*Proof.* Let us assume that $i$ preprepared $r$. There exist two mutually exclusive cases for request $r$. Either $i$ preprepared $r$ in some previous epoch $e' < e$ or $i$ prepares $r$ in $e$. We exhaustively show that in both cases, either $i$ has delivered $r$ or $i$ has un-marked $r$ from being preprepared.

1. $i$ preprepared $r$ in epoch $e' < e$. There are two possible cases.

   (a) $i$ delivered $r$ in $e'$. The lemma follows.

   (b) $i$ did not deliver $r$ in $e'$. In this case, since an epoch change occurred, $i$ resurrected $r$ at the end of $e'$ (Algorithm 4, lines 162-176), and, therefore, un-marked $r$ as preprepared.

2. $i$ preprepared $r$ in $e$. Then some node proposed $r$ in epoch $e$. There are two sub-cases to distinguish.

   (a) $i$ proposed $r$. This cannot happen by the statement of the lemma.

(b) Some other node $j$ proposed $r$. Here we distinguish again cases.

    i. $i$ already delivered $r$. The lemma follows.

    ii. $i$ did not deliver $r$. This means that the batch with $r$ is still considered in-flight by $i$ and, therefore, $i$ bucket re-distribution could not have happened at $i$ (Alg. 4, Line 68) since $j$ was assigned the bucket of $r$. Therefore, $i$ is not assigned the bucket of $r$. A contradiction to $i$ being assigned the bucket of $r$ (from the premise of the lemma). $\qquad\square$

**Lemma 9** (Liveness with Finitely Many Epochs). *In an execution with a finite number of epochs, if a correct client broadcasts request $r$, then some correct node eventually delivers $r$.*

*Proof.* Assume, by contradiction, that no correct node delivers $r$. This implies that no correct node delivers $r$ in the last epoch $e_{last}$.

By Lemma 2, $e_{last}$ is an infinite, stable epoch. Therefore all nodes, including $i$, are leaders. Since the epoch is infinite and the sequence numbers of the epoch are distributed to the leaders in a round robin way (Alg. 4, Line 179), $i$ will propose infinitely often.

By Lemma 6, and since the oldest request is always proposed first (Alg. 4, Line 179), $i$ will eventually have proposed all requests older than $r$ and $r$ will be the oldest request in $i$'s bucket queues.

Next time $i$ proposes a batch from $r$'s bucket we distinguish two cases.

1. $i$ has already proposed $r$ in some batch $b$ in $e_{last}$.

2. $i$ has not proposed $r$ in $e_{last}$.

In the second case, by Lemma 8, $r$ is not marked as pre-prepared at $i$. Since $r$ is the oldest request and $i$ is proposing from $r$'s bucket, $i$ will propose $r$ in its next batch $b$ (see also Sec 5.3, PRE-PREPARE paragraph).

Let $sn$ be the sequence number of batch $b$. From our contradiction statement, we have that $b$ will never be delivered. By Lemma 7, $b$ is the only batch that can be delivered with $sn$ in $e_{last}$. Therefore, no batch can be delivered with $sn$. This will trigger an epoch change timeout at all other correct nodes, causing an epoch change. A contradiction to $e_{last}$ being stable. $\qquad\square$

**Definition 1** (Preferred request). *Request $r$ is called preferred request in epoch $e$, if $r$ is the oldest request in the bucket queues of the primary node of epoch $e$, before the primary proposes its first batch in $e$.*

**Lemma 10.** *If all correct nodes perform an ungracious epoch change from $e$ to $e+1$ and the primary of $e+1$ is correct, then all correct nodes reliably deliver the epoch configuration of $e+1$.*

*Proof.* Let $p$ be the correct primary of $e+1$. As $p$ is correct, by the premise, $p$ participated in the ungracious epoch change sub-protocol. Since the PBFT view change protocol is part of the ungracious view change, $p$ sends a NEW-EPOCH message to all nodes. By the algorithm (Algorithm 4, line 150), $p$ reliably broadcasts the epoch configuration of $e+1$. Since all correct nodes participate in the ungracious view change, all correct nodes enter epoch $e+1$. By the properties of reliable broadcast, all correct nodes deliver the epoch configuration in $e+1$ (Algorithm 4, line 153). $\qquad\square$

**Lemma 11.** *There exists a time after GST, such that if each correct node reliably delivers the configuration of epoch $e$ after entering $e$ through an ungracious epoch change, and the primary of $e$ is correct, then all correct nodes commit $e$'s preferred request $r$.*

*Proof.* Let $p$ be the primary of epoch $e$, and $C$ the epoch configuration $p$ broadcasts for $e$. By the algorithm, the leaderset in $C$ does not contain all nodes (and thus SVS is disabled in $e$), as all correct nodes entered $e$ ungraciously. Since (by the premise) all correct nodes deliver $C$, all correct nodes will start participating in the common-case agreement protocol in epoch $e$. Let $t_f$ and $t_l$ be the time when, respectively, the first and last correct node does so.

By the algorithm, $p$ proposes $r$ immediately in some batch $B$ when entering epoch $e$, and thus at latest at $t_l$. Then, by Lemma 4, there exists a $\Delta$ such that all correct nodes commit $r$, as part of $B$, if none of them initiates a view change before $t_l + \Delta$.

Eventually, after finitely many ungracious epoch changes, where all correct nodes double their epoch change timeout values (as done in PBFT [24]), all correct nodes' epoch change timeout will be greater than $(t_l - t_f) + \Delta$. Then, even if a node $i$ enters epoch $e$ and immediately starts its timer at $t_f$, $i$ will not enter view change before $t_l + \Delta$ and, thus, all correct nodes will commit $r$ in epoch $e$. $\qquad\square$

**Lemma 12.** *There exists a time after GST, such that if all correct nodes perform an ungracious epoch change from $e$ to $e+1$, and the primary of $e+1$ is correct, then some correct node commits preferred request in $e+1$.*

*Proof.* Follows from Lemmas 10 and 11. □

**Lemma 13.** *In an execution with infinitely many epochs there exists an infinite number of pairs of consecutive epochs with correct primaries.*

*Proof.* Epoch primaries succeed each other in a round robin way across all the lexicographically ordered nodes of the system (see Sec. 4 and Sec. 2). Assume such pair of two consecutive epochs with correct primaries never exists after some epoch $e$. Then, in every full rotation across all $3f+1$ nodes after $e$, there exists an epoch with a faulty primary node between every two epochs with correct primaries, which implies the number of faulty nodes to be greater than $f$. A contradiction. □

**Lemma 14.** *There exists a time after GST, such that for any pair of consecutive epochs $e$ and $e+1$ with correct primaries $i$ and $j$ (respectively), some correct node commits at least one of the preferred requests in $e$ and $e+1$.*

*Proof.* Let $r_e$ (resp., $r'_e$) be preferred request in $e$ (resp., $e'$). For the epoch change from $e$ to $e+1$ there are two exhaustive possibilities.

1. *At least one correct node performs a gracious epoch change from $e$ to $e+1$.* Recall that Mir requires the primary of an epoch to be in the leaderset (Section 4). As $e$ graciously ends at at least one correct node, it follows from the specification of the gracious epoch change (Section 5), that at least one node commits all requests proposed in $e$.

   Since, by the protocol, the primary of $e$ is in the leaderset of $e$ and the correct primary always proposes the preferred request, at least one correct node commits the preferred request of $e$.

2. *No node performs a gracious epoch change.* By Lemma 12.

   □

(P5) **Liveness:** If a correct client broadcasts request $r$, then some correct node eventually delivers $r$.

*Proof.* We distinguish two cases:

1. In an execution with a finite number of epochs, Liveness follows from Lemma 9.

2. Consider now an execution with an infinite number of epochs. By Lemma 3, every correct node eventually receives $r$. Let $P$ be the set of all requests that some correct node received before it received $r$. After $r$ has been received by all correct nodes, following from Definition 1, if $r' \neq r$ is a preferred request, then $r' \in P$. By Lemma 14, however, all such requests $r'$ will eventually be committed by all correct nodes. Therefore, by Definition 1, unless $r$ is committed earlier by some correct node, $r$ will eventually become the preferred request of all epochs with correct primaries, and will be committed by some correct node by Lemma 14. Finally, by Lemma 5 $r$ will be delivered by some correct node.

   □

(P6) **In-order delivery:** If a correct node $i$ delivers some request $r$ with sequence number $sn$, then $i$ has delivered requests for each sequence number $sn'$, such that $sn' < sn$.

*Proof.* This property is trivially guaranteed by the way the protocol is designed. A correct node delivers a batch with sequence number $bsn$ only after it has delivered all batches with sequence numbers $bsn'$ with $bsn' < bsn$ (see Sec. 5.3, In-order delivery and Algorithm 4, line 119). Moreover, the request sequence numbers are assigned with a contiguous, monotonically increasing function (see Sec. 5.3, In-order delivery). In-order delivery follows. □

# 8 LTO: Optimization for large requests

When the system is network-bound (e.g., with large requests, such as those found in Hyperledger Fabric and/or on a WAN) the maximum throughput is driven by the amount of data each leader can send in a PRE-PREPARE message. However, data, i.e., request payload, is not critical for total order, as the nodes can establish total order on request hashes. While in many blockchain systems all nodes need data [1, 3], in others [11], ordering is separated from request execution and full payload replication across ordering nodes is unnecessary.

For such systems, Mir optionally boosts throughput using what we call *Light Total Order (LTO)* broadcast. LTO is defined in the same way as TO broadcast (Sec. 2) except that LTO requires property $P4$ (Totality) to hold for the hash of the request $H(r)$ (instead for request $r$).

P4 **Totality:** If a correct node delivers a request $r$ or a request hash $H(r)$, then every correct node eventually delivers $H(r)$.

On a high level, LTO follows a similar pattern as SVS (Sec. 5.6) and applies to Mir only in stable epoch. A leader only sends a full PRE-PREPARE message to a subset of $f+1$ *replica* nodes. To the remaining $2f$ *observer* nodes, the leader sends a lightweight PRE-PREPARE message where request payloads are replaced with their hashes.

Inside the Mir (and PBFT) common-case (Sec. 5.3) sub-protocol, before sending a COMMIT message, a node waits to receive at least $2f+1$ PREPARE messages, *such that $f+1$ of them are from the replica nodes*. This ensures that at least one correct (replica) node has the full payload.

LTO has minor impact on PBFT view change (Mir epoch change) as a new primary might have a hash of the batch (lightweight PRE-PREPARE) but not the full batch payload. To this end, our Mir-LTO makes the primary in this situation look for the payload at $f+1$ replicas, which is guaranteed not to block liveness after GST and with the correct primary.

| Max batch size | 2 MB (4000 requests) |
|---|---|
| Cut batch timeout | 500 ms ($n < 49$), 1s($n = 49$), 2s($n = 100$) |
| Max batches ephemeral epoch | 256 ($n \leq 16$), $16 * n$ ($n > 16$) |
| Bucket re-distribution period | 256 ($n \leq 16$), $16 * n$ ($n > 16$) |
| Buckets per leader ($m$) | 2 |
| Checkpoint period | 128 |
| Watermark window size | 256 |
| Parallel gRPC connections | 5 ($n = 4$), 3 ($n = 10$), 1 ($n > 10$) |
| Client signatures | 256-bit ECDSA |

Table 3: Mir configuration parameters used in evaluation

# 9 Evaluation

In this section, we report on experiments we conducted in scope of Mir performance evaluation, which aims at answering the following questions:
(Sec. 9.1) How does Mir scale on a WAN?
(Sec. 9.2) How does Mir scale in clusters?
(Sec. 9.3) What is the impact of optimizations (SVS, LTO) and bucket re-distribution and what are typical latencies of Mir?
(Sec. 9.4) What is the benefit of Mir duplication prevention?
(Sec. 9.5) How does Mir perform under faults and attacks (crash faults, censoring attacks, straggler attacks)?

**Experimental Setup.** Our evaluation consists of microbenchmarks of 500 byte requests, which correspond to average Bitcoin tx size [7]. These are representative of Mir performance, both absolute and relative to state of the art. We also evaluate Mir in WAN for larger 3500 byte requests, typical in Hyperledger Fabric [11] to better showcase the impact of available bandwidth on Mir.

We generate client requests by increasing the number of client processes and the request rate per client process, until the throughput is saturated. We report the throughput just below saturation. The client processes estimate which node $i$ has an active bucket for each of their requests and initially send each request only to nodes $i-1, \cdots, i+k$, where $k \leq f-1$, i.e., to $f+1$ nodes.

We compare Mir to a state-of-the-art PBFT implementation optimized for multi-cores [16]. For fair comparison, we use the Mir codebase tuned to closely follow the PBFT implementation of [16] (see also Section 4, paragraph "Generalization of PBFT and Emulation of Other BFT Protocols") hardened, as Mir, to implement Aardvark [25] (client signatures instead of MAC vectors and separate network interface for client requests and protocol messages). As another baseline, we compare the common case performance of Chain, an optimistic sub-protocol of the Aliph BFT protocol [13] with linear common-case message complexity, which is known to be near throughput-optimal in clusters, although it is not robust and needs to be abandoned in case of faults [13]. In this sense, Chain is not a competitor to Mir, but rather an upper bound on performance in a cluster. Our PBFT and Chain implementations have the same batching mechanism as Mir (see Sec. 5.3, PRE-PREPARE paragraph). Moreover, PBFT and Chain are always given best possible setups, i.e., PBFT leader is always placed in a node that has most effective bandwidth and Chain spans the path with the smallest latency. We further compare to HotStuff [61] (a recent, popular, $O(n)$ common-case message complexity BFT protocol) and Honeybadger [51] using their open source implementations [6]. We present comparison to HotStuff separately, due to its implementation specifics. We allow Honeybadger an advantage with using 250 byte requests, as its open

---

[6] https://github.com/hot-stuff/libhotstuff at commit 978f39f... and https://github.com/initc3/HoneyBadgerBFT-Python

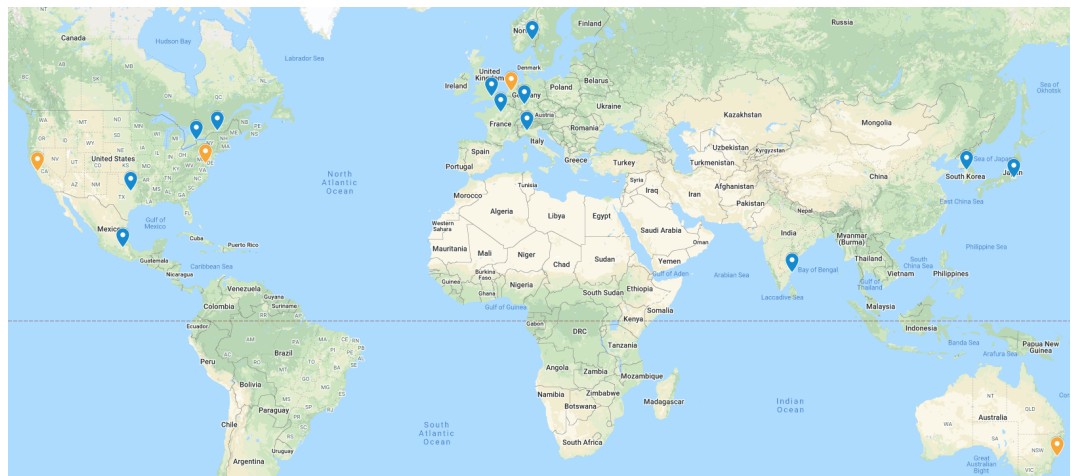

Figure 4: Distribution of the 16 datacenters for WAN deployment. Yellow pins indicate the $n = 4$ deployment.

source implementation is fixed to this request size. We do not compare to unavailable (e.g., Hashgraph [45], Red Belly [26], RCC [38], OMADA [30]) or unmaintained (BFT-Mencius [52]) protocols. We, however, demonstrate the expected effective throughput of Hashgraph, Red-Belly, BFT-Mencius, RCC, and OMADA under request duplication, by "switching off" request duplication prevention in Mir, see Sec. 9.4. We further do not compare to single leader protocols faithfully approximated by PBFT (e.g., BFT-SMaRt [17], Spinning [57] or by Hot-Stuff (e.g., SBFT [36]) or those that report considerably worse performance than Mir (e.g., Algorand [35]).

We use virtual machines on IBM Cloud, with 32 x 2.0 GHz VCPUs and 32GB RAM, equipped with 1Gbps networking and limited to that value for experiment repeatability, due to non-uniform bandwidth overprovisioning we sometimes experienced. Table 3 shows the used Mir configuration parameters. Unless stated otherwise, Mir uses the signature verification sharding optimization.

## 9.1 Scalability on a WAN

To evaluate Mir scalability, we ran it with up to $n = 100$ nodes on a WAN setup spanning 16 distinct datacenters across Europe, America, Australia, and Asia. Beyond $n = 16$, we collocate nodes across already used datacenters. Our 4-node experiments spread over all 4 mentioned continents. Client machines are also uniformly distributed across the 16 datacenters. Figure 4 shows the datacenter distribution.

Figure 5a depicts the common-case (failure-free) stable epoch performance of Mir, compared to that of PBFT,

Chain, and Honeybadger. We observe that PBFT throughput decays rapidly, following an $O(1/n)$ function and scales very poorly. Chain scales better, sustaining 20k req/s, but is limited by the bandwidth of the "weakest link", i.e., a TCP connection with lowest bandwidth across all links between consecutive nodes. In fact, Chain throughput improves with up to $n = 16$ nodes since adding more datacenters yields a path with nodes physically closer to each other, improving the per TCP connection bandwidth. Compared to Honeybadger, Mir retains much higher throughput, even though: (i) Honeybadger request size is smaller (250 bytes vs 500 bytes), and (ii) Honeybadger batches are significantly larger (up to 500K requests in our evaluation). This is due to the fact that Honeybadger is computationally bound by $O(n^2)$ threshold signatures verification and on top of that the verification of the signatures is done sequentially. Honeybadger's throughput also suffers from request duplication (on average $1/3$ duplicate requests per batch), since the nodes choose the requests they add in their batches at random. Moreover, we report on Honeybadger latency, which is in the order of minutes (partly due to the large number of requests per batch and partly due to heavy computation), significantly higher than that of Mir. In our evaluation we could not increase the batch size as much as in the evaluation in [51], especially with increasing the number of nodes beyond 16, due to memory exhaustion issues. Finally, in our evaluation PBFT outperforms Honeybadger (unlike in [51]), as our implementation of PBFT leverages the parallelism of Mir codebase.

Mir dominates other protocols, delivering 82.5k, roughly 4x the throughput of Chain, with $n = 4$. The

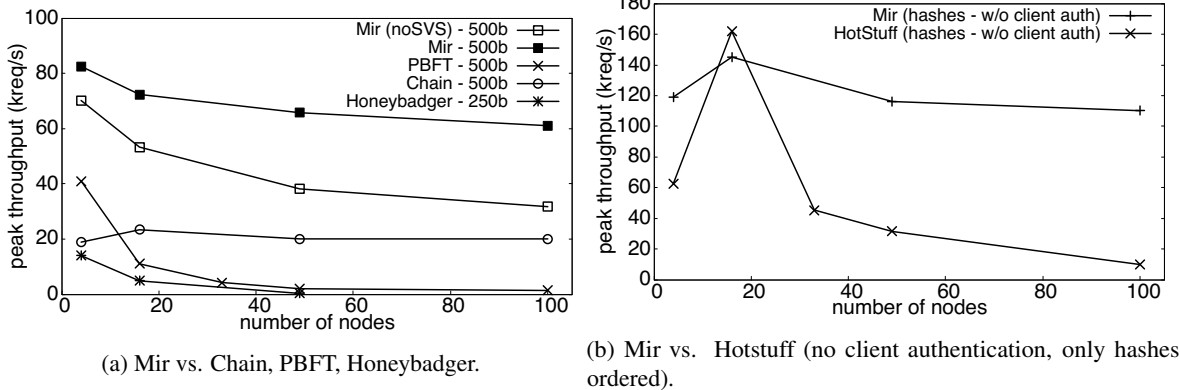

(a) Mir vs. Chain, PBFT, Honeybadger.

(b) Mir vs. Hotstuff (no client authentication, only hashes ordered).

Figure 5: WAN scalability experiments.

improvement over Chain throughput is thanks to Mir opening multiple TCP connections between each pair of nodes and therefore utilizing more effectively the available bandwidth to all nodes. Multiple TCP connections is a low-level optimization which is empirically found to increase the effective bandwidth between two nodes. Importantly, Chain cannot support multiple TCP connections, since this might violate the FIFO channels between nodes, on which Chain, unlike Mir, relies. With $n = 100$, Mir maintains more than 60k req/s (3x Chain throughput). Even without the signature verification sharding optimization ("Mir (noSVS)") Mir significantly outperforms other protocols, delivering with $n = 4$ 70.2k req/s (3.5x Chain throughput) while reaching 31.7k req/s with $n = 100$ (1.5x Chain throughput).

**Comparison to HotStuff in WANs.** We present the comparison of Mir to the HotStuff [61] leader-based protocol separately, in Figure 5b. Despite HotStuff specifying that the leader disseminates the request payload [61], the available HotStuff implementation orders only hashes of requests, relying optimistically on clients for payload dissemination. This approach is vulnerable to liveness/performance attacks from malicious clients which can be easily mounted by clients not sending the requests to all nodes (an attack which the HotStuff version we evaluated does not address). Besides, the evaluated HotStuff implementation did not authenticate clients at all (which jeopardizes Validity).

For these reasons and for a fair comparison, we perform an experiment with: 1) disabled Mir client authentication (i.e,. client signature verification) and 2) with leaders disseminating payload hashes (relying on clients to disseminate payload as in HotStuff). We also increase batch sizes in HotStuff as much as needed, resulting in

up to 32K requests per batch, to saturate the system.

We observe that HotStuff offers about 2x lower throughput than Mir with $n = 4$ nodes bounded by the number of available network connections, whereas Mir uses multiple connections among pairs of nodes. As $n$ and number of network connections from the leader grow, HotStuff throughput first grows until the network at the leader is saturated (with $n = 16$ HotStuff performs about 10% better than Mir). However, as leader bandwidth becomes the bottleneck even with hash-only ordering, HotStuff's $O(n^{-1})$ network-bound scalability starts to show with $n > 16$, while Mir continues to scale well and is only computationally bounded by the implementation. With 100 nodes, Mir orders 110k hashes per second, compared to roughly 10k hashes per second throughput of HotStuff.

**Experiments with 3500-byte payload.** With small request payload size (500 bytes), CPU overhead related to signature verification is the primary bottleneck. It is therefore interesting to evaluate the impact on performance with larger requests. Intuitively, with larger requests, we would be able to stress the 1Gbps WAN bottleneck of our evaluation testbed. Moreover, large requests are not only of theoretical importance, some prominent blockchain systems feature relatively large transaction sizes. For instance, minimum size transaction in Hyperledger Fabric is about 3.5kbytes [11].

Therefore, we conducted additional WAN experiments with 3500 bytes request size.

For large requests, where network bandwidth is the bottleneck, throughput of Mir (with no SVS) reduces to 7k req/s with 100 nodes, with a drop from 28.3k req/s with $n = 4$ nodes, see Figure 6. We attribute this in part to the heterogeneity of VMs across datacenters (despite

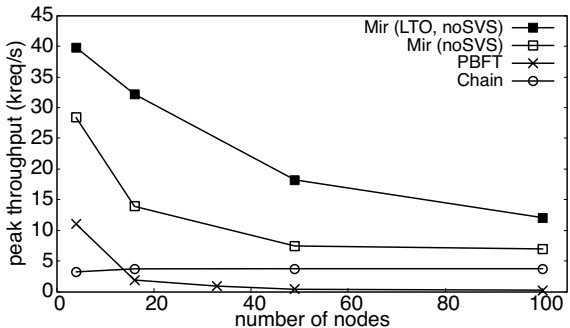

Figure 6: WAN scalability experiment with large payload (3500 bytes).

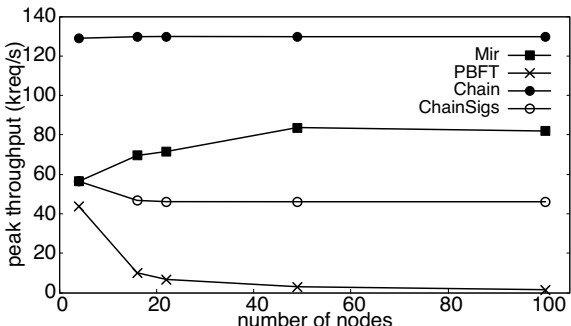

Figure 7: Throughput performance of Mir compared to Chain and PBFT in a single datacenter (500 bytes).

the identical specifications) and, most importantly, to the non-uniform partition of the available uplink bandwidth. Nevertheless, Mir delivers the best performance of all protocols to date with 100 nodes on a WAN, even compared to very optimistic protocols such as Chain, which delivers consistent throughput of about 4.5k req/s regardless of number of nodes. Mir is, hence, the first robust BFT protocol which could be used as an ordering service in Fabric with $n = 100$ nodes, without making ordering service a bottleneck (validation in Fabric is currently capped at less than 4k transactions per second [11]).

In addition to Mir (with no SVS), Chain and PBFT, Figure 6 also shows an experimental variant of Mir which implements what we call *Light Total Order (LTO)* broadcast, instead of full TOB (labeled 'Mir (LTO, noSVS)'). As described in Section 8, LTO is an optimization, counterpart of SVS, to help alleviate network bottlenecks in TOB. In short, LTO broadcast is identical to TOB, except that it provides partial data availability guaranteeing the delivery of the *payload* of every request to *at least one* correct node. This entails replicating batch payload to $f + 1$ nodes in stable epochs, compared to all nodes without LTO. Other correct nodes get and agree on the order of cryptographic hashes of requests, which is the basis for maintaining other TOB properties.

LTO boosts throughput of Mir to 40k 3500-byte req/s with $n = 4$ nodes (roughly 40% throughput improvement over Mir) and maintains about 12.5k req/s with $n = 100$ nodes (70% throughput improvement over Mir).

## 9.2 Scalability in a Cluster/Datacenter

Figure 7 depicts fault-free performance in a single datacenter with up to $n = 100$ nodes. Mir reaches 64% of Chain's peak throughput (83k req/s vs 130k req/s). This difference is due to a difference in client authentica-

tion: Mir verifies clients' signatures, whereas Chain uses vectors of MACs to authenticate a request to $f + 1$ replicas (these are vulnerable to "faulty client" attacks [25]). Recall that Chain is not a robust protocol itself, but an optimistic sub-protocol of the Aliph protocol [13]. Indeed, as soon as we add clients' signatures to Chain towards a robust version of Chain (denoted by ChainSigs in Fig. 7), Chain's throughput drops below that of Mir. Mir maintains more than 80k req/s throughput, significantly outperforming PBFT.

## 9.3 Impact of optimizations and bucket re-distribution

Fig. 8 shows the average latency and throughput of different flavors of Mir in fault-free executions using $n = 16$ nodes. We also show the performance of Chain and PBFT as a reference. Nodes are distributed over 16 distinct datacenters across the world.

Mir without signature verification sharding ("Mir (noSVS)" in Fig. 8) saturates at roughly 53k req/s (resp. 12.3k req/s for large requests). To evaluate the overhead of bucket re-distribution, we compare Mir without signature verification sharding to running parallel PBFT instances, sharing only a common checkpoint mechanism and watermark window ("Parallel PBFT"). For a fair comparison, the parallel PBFT instances implement the same mechanisms as Mir (signatures instead of MACs, separate network interface for clients and parallelized network access — see Section 5.10). Moreover, for the parallel PBFT instances, we evaluate a workload of unique client requests to remove the negative impact of duplication. We observe an approximate overhead of about 3% (resp. 9.5%) for adding bucket re-destribution. Importantly, the parallel PBFT instances protocol is not live, unless the client can resubmit their request (poten-

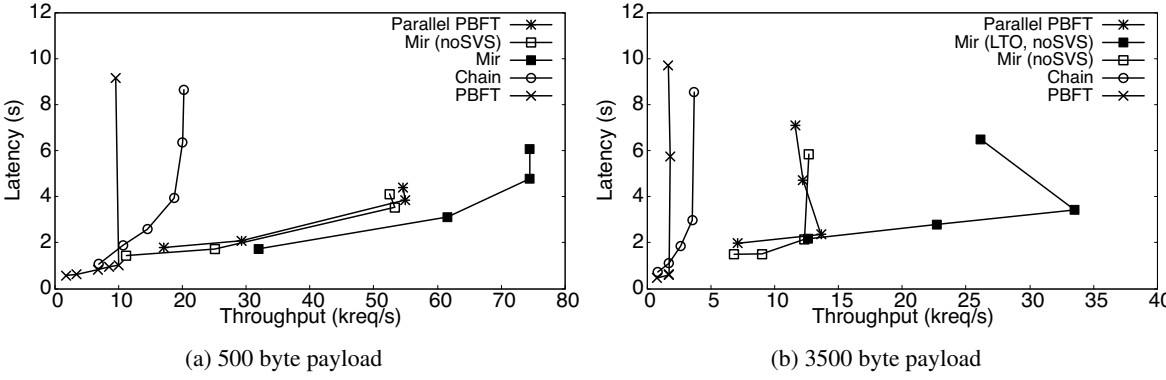

(a) 500 byte payload       (b) 3500 byte payload

Figure 8: Impact of bucket re-distribution (Mir vs parallel PBFT instances) and Mir optimizations on a WAN with n=16 nodes.

tially after a timeout, as done by BFT-Mencius [52] or RCC [38], substantially increasing latency) to at least $f$ other nodes. We evaluate the impact of submitting $f + 1$ or more requests in parallel without duplication prevention in the next section (Section 9.4).

The small penalty of robust bucket re-distribution is more than compensated for by signature sharding which boosts Mir throughput to 74k req/s (resp. to 33.5k req/s with LTO).

All variants of Mir maintain roughly from 1–2s latency at relatively low load, to 3–5s latency close to saturation. PBFT latency is lower at 600–800 ms, yet PBFT saturates under very low load compared to Mir. We measured latency by: (1) synchronizing clocks between a client and a node belonging to the same datacenter with NTP, (2) deducting request timestamp at a client from commit timestamp at a node, (3) averaging across all requests (and, consequently, all datacenters).

## 9.4 Benefits of Duplication Prevention

In this section we examine the impact of duplicate requests to *goodput*, i.e., throughput of unique requests. In Fig. 9 we compare the performance of Mir (noSVS) to a version of Mir where the leaders do not partition requests in buckets, but rather add in batches all their available requests, following what other parallel leader protocols do. These protocols include Hashgraph [45], Red Belly [26], BFT-Mencius [52], OMADA [30] and RCC [38] .

We examine the impact of duplicates in two scenarios, (1) where clients submit their requests to $f + 1$ nodes — intuitively, this is the minimum number of nodes to which a client must submit a request in any BFT protocol that

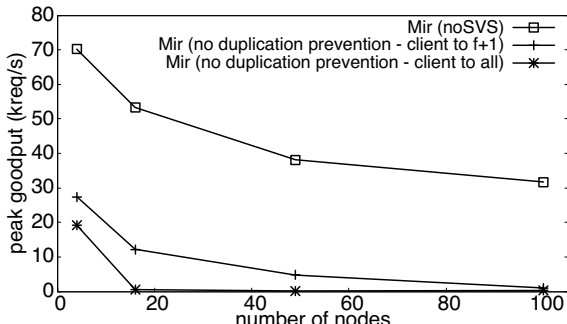

Figure 9: The impact of duplication prevention on a WAN with $n = 16$.

ensures liveness (due to possible censoring by f nodes), and (2) where clients submit their requests to all nodes.

The impact is a significant performance penalty of 61% (resp. 72%) reduction in goodput compared to Mir (noSVS) in the first (resp. second) scenario on $n = 4$ nodes. This reaches as much as 97% (resp. 99%) with $n = 100$, demonstrating $O(n^{-1})$ goodput scalability in protocols with duplication.

## 9.5 Performance Under Faults

**Leader Crash Faults.** Figure 10 shows throughput as a function of time when one and two leaders fail simultaneously. We run this experiment in a WAN setting with 16 nodes, and trigger a view change if an expected batch is not delivered with fixed timeouts of 20 seconds. With one leader failure, a view change is triggered and the system immediately transitions to a configuration with 15 leaders. When two failures occur simultaneously, the

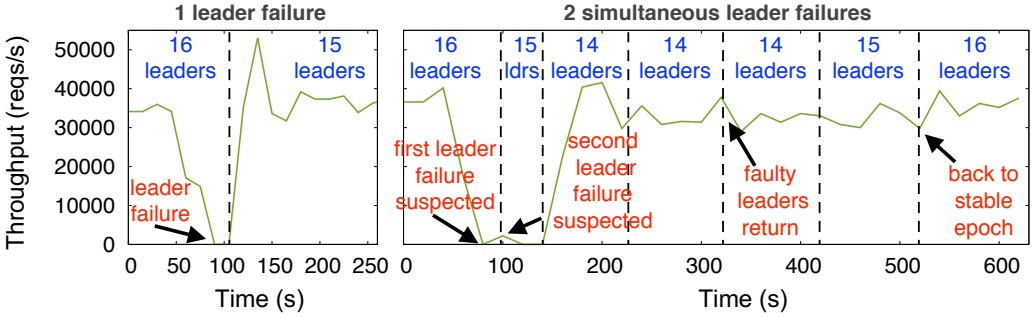

Figure 10: Performance under crash faults.

first view changes takes the system to a configuration with 15 leaders. The first few batches are delivered in this configuration, but, since one of the 15 leaders has failed, a second view change is triggered that takes the system to a configuration with 14 leaders, from which execution can continue normally. In this scenario, the figure also depicts the evolution of the leaderset in case the failed nodes recover: within three epochs, the system is in a stable state with 16 leaders again.

We can observe that gracious epoch changes are seamless in Mir (these occur from second 141 onwards in the experiment with 2 faults), whereas ungracious epoch changes (when throughput temporarily drops to 0) last approximately one epoch change timeout.

**Request Censoring (Byzantine Leaders Dropping Requests).** In this experiment we emulate Byzantine behavior by having an increasing number (from 0 to $f = 5$) of Byzantine leaders dropping (censoring) requests in our 16-node WAN setup. Fig. 11a shows that mean latency remains below 4.6s (resp., 2.2s) when Byzantine leaders drop 100% (resp. 25%) of the requests they receive. Tail latencies (95th percentile) remain below 16s (resp., 7s). Fig. 11b shows the distribution and CDF of latency with 5 Byzantine leaders censoring 100% of requests. When clients send requests to all nodes, we observe a drop of up to 15% for mean and 18% for tail latencies. In an experiment with bucket re-distribution period of 128 batches (not depicted) we observe a decrease of up to 44% for mean and 49% for tail latencies. This introduces, though, a penalty of approximately 10% in peak throughput.

**Stragglers (Byzantine Leaders Delaying Proposals).** In this experiment we evaluate Mir resistance to stragglers. Stragglers delay the batches they lead and propose empty batches. In detail, if the epoch change timer expiration duration is $D$, a straggler delays the proposal as much as possible without triggering any batch timeouts. The upper bound on the straggler delay is thus $D$.

In Mir, with multiple leaders proposing and committing batches independently, a single straggler can only impose a total delay of at most $D$ *once per epoch* without being detected, as compared to once per sequence number, in single-leader protocols. The key to Mir straggler resistance is that each sequence number $sn$ has its own epoch change timeout which a correct node starts as soon as it commits $sn - 1$ (Alg. 4, Line 116). Moreover, the sequence numbers are assigned to leaders in a round robin way and therefore a single straggler does not control the proposal of contiguous sequence numbers. Therefore, batches committed by correct nodes will trigger timers for the straggler's batches independently leading to those timers running mostly in parallel. In our implementation, the next epoch primary will suspect as faulty the node who was responsible for the sequence number whose timer expired, in this case the straggler, and remove this node from the next leaderset. The straggler is re-added back to the leaderset only once it becomes epoch primary.

We perform both WAN and LAN experiments with $n = 16$ nodes, starting from a stable epoch. The load is set at about 25-30% peak throughput (corresponding to roughly 25k req/s). Epoch change timeout is set to 20s and ephemeral epoch length to 256 batches. We run our experiment until the straggler is removed from and re-added to the leaderset. .

On WAN, fault-free throughput gives a baseline of 24.8k req/s. With a single Byzantine straggler leader delaying each of its batches by 15s, the average throughput is 18k req/s (penalty of 25% over the baseline). The straggler is always detected and removed from the leaderset almost immediately.

On LAN, baseline throughput without faults is 28.1k req/s. For reference, Mir latency in LAN is in milliseconds. We set straggler delay to 2 seconds (while keeping epoch change timeout to 20s – a value verybig for LAN) to keep the straggler longer in the leaderset. This

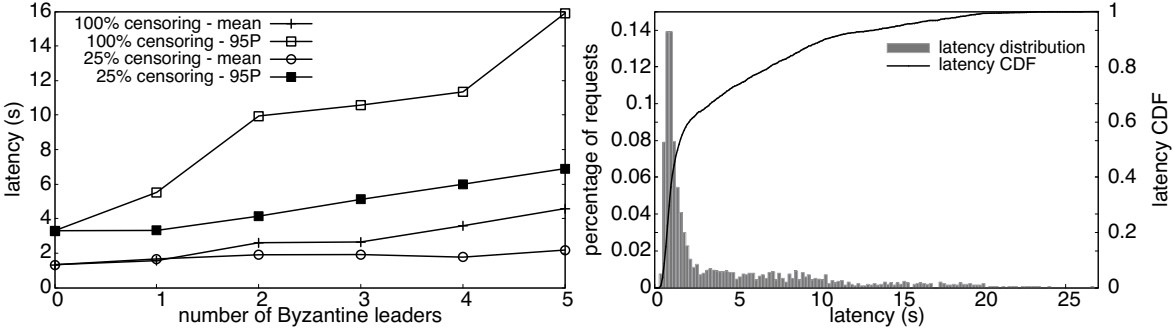

(a) Mean and tail latencies (95%) for increasing number of Byzantine leaders that drop 25% or 100% of their requests.

(b) Latency distribution and CDF with $f = 5$ Byzantine leaders censoring 100% of requests.

Figure 11: Latency under the request censoring attack.

time, the straggler remains in the leaderset for over 600 sequence numbers, after which it is removed from the leaderset. In this case, we measure average throughput of 15.7k req/s in the entire execution (a penalty of 44%).

To put these numbers into perspective, a single-leader Aardvark [25] suffers a 90% performance penalty with a straggler primary on a LAN delaying batches for 10ms. We conclude that Mir has very good performance in presence of stragglers, even with simple fixed epoch change timeouts. Future optimizations of Mir Byzantine node detection are possible, following the approaches of Aardvark [25] and RBFT [12].

## 10 Related Work

The seminal PBFT [24] protocol sparked intensive research on BFT. PBFT itself has a single-leader network bottleneck and does not scale well with the number of nodes. Mir generalizes PBFT and removes this bottleneck with a multi-leader approach, enforcing a robust request duplication prevention. Request duplication elimination is simple in PBFT and other single-leader protocols, where this is the task of the leader.

Aardvark [25] was one of the first BFT protocols, along with [10,12,57], to point out the importance of BFT protocol *robustness*, i.e., guaranteed liveness and reasonable performance in presence of active denial of service and performance attacks. In practice, Aardvark is a hardened PBFT protocol that uses clients' signatures, regular periodic view-changes (rotating primary), and resource isolation using separate NICs for separating client-to-node from node-to-node traffic. Mir implements all of these and is thus robust in the Aardvark sense. Beyond Aardvark features, Mir is the first protocol to combine robustness with multiple leaders, preventing request duplication performance attacks, enabling Mir's excellent performance.

The first replication protocol to propose the use of multiple parallel leaders was Mencius [50]. Mencius is a crash-tolerant Paxos-style [46] protocol that leverages multiple leaders to reduce the latency of replication on WANs, an approach later followed by other crash-tolerant protocols (e.g., EPaxos [53]). The approach was extended to the BFT context by BFT-Mencius [52]. Mencius and BFT-Mencius are geared towards optimizing latency and shard clients' requests by mapping a client to a closest node. If the clients' request is not delivered within a timeout period, the client retrasnmits its request to another node. However, this technique has several reciprocations which BFT-Mencius does not handle.

- The client cannot distinguish a faulty leader from a slow leader, therefore it is infeasible for the client to choose a timeout that prevents from introducing duplicate requests to the system.

- If the client has a conservative timeout, the client might suffer very long latency, especially in the case of cascading faults (or Byzantine nodes who accept the request pretending to be correct).

- Worse, if a client is malicious, the client can retransmit its request to other nodes exposing a vulnerability to request duplication attacks.

As illustrated in our evaluation (Sec. 9.4), malicious clients can severely impact the throughput of such a scheme, by sending their requests to multiple or all nodes. Unlike in a regular DoS attack, these clients cannot be

naively declared Byzantine or rate-limited, as such request traffic may be needed by correct clients to deal with Byzantine leaders dropping requests (request censoring attack) or to optimize the latency of a BFT protocol. Unlike BFT-Mencius, Mir maps clients' requests to buckets which are then assigned to nodes, similarly to consistent hashing [39]. Mir further redistributes bucket assignment in time to enforce robustness to request censoring. Unlike Mencius, EPaxos and BFT-Mencius, Mir does not optimize for latency in the best case, paying a small price as it does not assign clients to the closest nodes.

Recent BFT protocols proposed in the blockchain context (Hashgraph [45], RedBelly [26], and OMADA [30]) that exhibit a multi-leader flavor, also do not address request duplication. In particular RedBelly and OMADA address request duplication similarly to BFT-Mencius, suffering the same disadvantages. Notably, RedBelly introduces a similar mechanism to SVS for improving performance.

Hashgraph suggests charging fees for duplicate requests. This can, however, unfairly penalizes correct clients whose requests are delayed or censored. Furthermore, unlike Mir, Hashgraph invents a new BFT protocol from scratch which is a highly error-prone and tedious process [13]. In contrast, Mir follows an evolutionary rather then revolutionary design approach to a multi-leader protocol, building upon proven PBFT/Aardvark algorithmic and system constructs, considerably simplifying the reasoning about Mir correctness.

Parallel to this work, RCC [38] introduced a wait-free paradigm for multiplexing single-leader protocol instances. While the wait-free design reduces the impact of failure recovery on throughput, RCC throughput may degrade with malicious clients or network asynchrony. In particular, RCC uses consensus among replicas to allow the client to pick an arbitrary single instance *per round* of agreement. Besides allowing faulty clients to overload a single replica, there are executions where even a correct client, under asynchrony, sends a request to some instance and then asks to switch instances, causing both the "old" and the "new" instance to *propose* the same request (albeit in different rounds), wasting bandwidth. This can be generalized to more than two instances. RCC suggests a synchronization mechanism to prevent a client from subscribing to multiple instances. Mir completely prevents multiple proposals of the same request. Moreover, Mir aims to mitigate uneven request distribution using a hash function for leader assignment, along with the client watermark window mechanism. Finally, in RCC, delayed messages or a faulty client can cause unnecessary request re-transmissions. Mir relies on the

clients submitting the request to enough nodes and on the bucket re-assignment mechanism to guarantee liveness, preventing such re-transmissions by design. While it seems there is a latency - throughput trade-off, Mir evaluation (Fig. 11) shows that, even under heavy censoring, request delay is limited to the order of seconds, a latency that seems comparable to a timeout that would be necessary to preserve liveness with the request re-transmission or instance-switching mechanism.

Two recent protocols, HotStuff [61] and SBFT [36], are leader-based protocols that improve on PBFT's quadratic common-case message complexity and require a linear ($O(n)$) number of messages in the common case. HotStuff is optimized for throughput and features $O(n)$ messages in view change as well (SBFT requires $O(n^2)$ messages in view change). While Mir's approach of multiplexing PBFT instances and SBFT/HotStuff improvements over PBFT appear largely orthogonal, our experiments show that Mir multi-leader approach scales better than HotStuff, which is a single-leader protocol. Namely, even though PBFT/Mir have quadratic common-case message complexity, these messages are load balanced across $n$ nodes, yielding $O(n)$ messages at a *bottleneck replica*, just like HotStuff/SBFT. Our experiments also showed that HotStuff retains the downside of other single-leader protocols, i.e., bottlenecks related to leader sending all proposals, yielding an unfavorable $O(n^{-1})$ throughput scalability trend. An unimplemented HotStuff variant, called ChainedHotStuff [61], suggests having different leaders piggyback their batches on other protocol common-case messages. As Hotstuff has 4 common case phases, this allows up to 4 "chained" leaders in ChainedHotStuff regardless of the total number of nodes, which is less efficient than Mir which allows up to $n$ parallel leaders. In future, it would be very interesting to combine the two approaches, $O(n)$ common case message complexity and parallel leaders, by implementing Mir variants based on HotStuff/SBFT instead of PBFT.

Tendermint [19] authors, on the other hand, realize that a leader (proposer) who is responsible for disseminating all transactions quickly becomes a bottleneck. To address this, while maintaining a single rotating leader protocol, they offload the transaction dissemination to an underlying gossip protocol. The leader adds in a block proposal for a certain height only the Merkle root hash of the transactions. However, liveness cannot be guaranteed, unless the rest of the nodes (validators), wait for all transactions from the gossip protocol before voting for the proposal. Moreover, the block for the next height cannot be proposed before agreeing on a block for the current height. This makes Tendermint latency bound, i.e., it's

performance depends on the network latency. Thus, the only way to increase throughput is via aggressive batching which, however, increases end-to-end latency. Even worse, due to asynchrony, it may take multiple rounds of proposals per height. Hashgraph [45] is an example of a protocol that runs multiple gossip instances. However, running multiple gossip instances is not more efficient in terms of message complexity, when compared to Mir. In particular, in Mir request dissemination has a *per node* $O(n)$ message complexity, while $O(n)$ gossip instances result in $O(n)$ times the underlying gossip protocol complexity per node.

*Optimistic* BFT protocols [13, 44] have been shown to be very efficient on a small scale in clusters. In particular, Aliph [13] is a combination of Chain crash-tolerant replication [56] ported to BFT and backed by PBFT/Aardvark outside the optimistic case where all nodes are correct. We demonstrated that Mir holds its ground with Chain in clusters and it considerably outperforms it in WANs. Nevertheless, Mir remains compatible with the modular approach to building optimistic BFT protocols of [13], where Mir can be used as a robust and high-performance backup protocol. Zyzzyva [44] is an optimistic leader-based protocol that optimizes for latency. While we chose to implement Mir based on PBFT, Mir variants based on Zyzzyva's latency-efficient communication pattern are conceivable.

Eventually synchronous BFT protocols, to which Mir belongs, circumvent the FLP consensus impossibility result [32] by assuming eventual synchrony. These protocols, Mir included, guarantee safety despite asynchrony but rely on eventual synchrony to provide liveness. Alternatively, probabilistic BFT protocols such as Honeybadger [51] and BEAT [28] provide both safety and liveness, except with negligible probability, in purely asynchronous networks. By comparing Honeybadger and Mir, we showed that this comes as a trade-off, as Mir significantly outperforms Honeybadger, even though both protocols target the same deployment setting (up to 100 nodes in a WAN). Notably, the authors realize the importance of duplicate elimination, targeting "mostly disjoint sets of transactions" in HoneyBadger's proposals. They suggest that each leader randomly samples the received and yet unproposed requests. While this approach would result to no duplicates on expectation with a sufficiently large pending request buffer, in practice, unless the system is deep in saturation, the request buffer does not contain significantly more requests than the next batch. Therefore, if the request buffers of multiple leaders contain duplicate requests, the leaders will include them in their respective batches. Indeed, in our Honeybadger

evaluation we observed that goodput (effective throughput) was roughly only 20% of the nominal throughput. BEAT suggests some optimizations over Honeybadger without significantly outperforming the former.

As blockchains brought an arms-race to BFT protocol scalability [58], many proposals focus on large, Bitcoin-like scale, with thousands or tens of thousands of nodes [31, 35]. In particular, Algorand [35] is a recent BFT protocol that deals with BFT agreement in populations of thousands of nodes, by relying on a verifiable random function to select a committee in the order of hundred(s) of nodes. Algorand then runs a smaller scale agreement protocol inside a committee. We foresee Mir being a candidate for this "in-committee" protocol inside systems such as Algorand as well as in other blockchain systems that effectively restrict voting to a smaller group of nodes, as is the case in Proof of Stake proposals [20]. In addition, Mir is particularly interesting to permissioned blockchains, such as Hyperledger Fabric [11].

ByzCoin [42] scales PBFT for permissionless blockchains by building PBFT atop of CoSi [55], a collective signing protocol that efficiently aggregates hundreds or thousands of signatures. Moreover, it adopts ideas from PoW based Bitcoin-NG [31] to decouple transaction verification from block mining. This approach is orthogonal to that of Mir and variants of Byzcoin with Mir instead of PBFT are interesting for future work.

Stellar [48] uses SCP, a Byzantine agreement protocol with asymmetric quorums and trust assumptions targeting payment networks, which targets similar network sizes as Mir. Asymmetric quorums of SCP modify trust assumptions and the liveness guarantees of traditional BFT protocols, with [41] showing liveness violation with failures of only two specific nodes in a production configuration of Stellar. We show it is possible to obtain high throughput and low latencies while maintaining the strong guarantees of BFT protocols with classical (symmetric) quorums and trust assumptions.

*Sharding* protocols [43, 49] partition transaction verification into independent shards. Mir is complementary to such protocols as they either require ordering within a shard or total ordering of the shards. Monoxide [59] also uses sharding to increase throughput, but provides weaker guarantees (eventual atomicity across shards). Moreover, Monoxide's scalability heavily depends on transaction payload semantics.

Finally, our work is already generating considerable traction, with recent follow-up works attempting to extend our approach (e.g., [14, 33, 37]), however without yet provably improving Mir performance.

## 11 Conclusions

This paper presented Mir, a high-throughput robust BFT protocol for decentralized networks. Mir is the first BFT protocol that uses multiple parallel leaders thwarting both censoring attacks and request duplication performance attacks. In combination with reducing CPU overhead through the "signature verification sharding" optimization, this allows Mir to achieve unprecedented throughput at scale even on a wide area network, outperforming state-of-the-art protocols.

The main insight behind Mir is multiplexing multiple parallel instances of the PBFT protocol into a single totally ordered log, while preventing duplicate request proposals by partitioning the request hash space and assigning each subset to a different leader. Mir prevents request censoring attacks by periodically changing this assignment to guarantee that each request is eventually assigned to a correct leader. Being based on the well understood and thoroughly scrutinized PBFT makes it is easy to reason about Mir's correctness.

## Acknowledgements

We thank Jason Yellick for his very insightful comments and suggestions.

## Artifacts

The code described in this paper and used to produce experimental results is available open source at the following URL https://github.com/hyperledger-labs/mirbft/tree/research.

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
