# OpenReview forum: "Solution: Mir-BFT: Scalable and Robust BFT for Decentralized Networks"
_JSYS/2021/May_Papers — JSYS May 21_

### Official Review · Reviewer_Mbij · 2021-05-27
**Review of the anonymous submission on SRB, a BFT total order broadcast protocol**

**Decision:**

Strong accept: excellent paper that will help the community

**Review:**


The paper presents a BFT total order broadcast system targeted for the support of blockchain systems, particularly for permissioned systems and as a module for agreement in Proof-of-Stake permissionless systems. While this is a crowded area of research, I believe the paper and the system it describes, SRB, presents an interesting combination of technology and delivers a system that improves performance over competing approaches. The target environment is for planet scale networks, and was evaluated up to 100 nodes. While it excels in WAN settings, the protocol also shows decent performance in LAN settings, albeit with lower throughput wrt the competing best algorithm.

The design is based on the classic PBFT protocol and makes possible the use of parallel leaders and to distribute the verification of client signatures. Correctness is based on the correctness of PBFT core algorithms, which brings some confidence on the properties given the high complexity of this kind of systems and the difficulty of evaluation for manual proofs, as is the case here. Even so, most arguments look sound and decisions taken looks rooted on solid grounds. I also appreciated that the protocol generalizes PBFT and allows emulation of other protocols. The text itself is well written and I only detected minor issues that should be simple to correct. While, I am not a specialist in BFT consensus, I am familiar with the concept, with approaches to crash fault consensus (non BFT), and some of the mentioned BFT designs and blockchains.

Bellow I list some more detailed comments on the paper content.

Page 2:

"This paper presents SRB (or, simply, SRB) ..." Show acronym in full.

Page 3:

System model assumes unique node identities and mapping into a [0,n-1] integer namespace. This is a static assumption, so it might merit some discussion on if some dynamicity on participating server nodes can be handled.

Page 5:

Epoch leaders, "i adds itself to the leader set of epoch e′ and removes one node (not itself) for each epoch between e and e′". Please explain why one  is removed per each epoch in between. If this is not an open choice and is required for correctness, maybe give some explanation here.

Page 6:

Please increase the size of Fig 2 to make it more readable. I believe there is no page limit in JSys.

Page 7:

"H(t $\parallel$ c)". Unless it is usual to use $\parallel$ in this setting, please consider a different symbol for concatenation, e.g. $t\cdot c$

Page 8:

"However, such such inconsistent epoch" Remove such

Page 9:

"information about committed batches.." Remove .

Is the "$⟨HELLO, ne_i , c_i , b_i ⟩$" not signed?
Can the reply to HELLO messages create excessive traffic if Byzantine nodes keep issuing HELLOs to induce replies from other nodes?

You state that "While details of membership reconfiguration are outside of the scope of this paper, we briefly describe how SRB deals with adding/removing clients and nodes." but recall this is a journal version with plenty of space.

Algorithm 2:

Lines 12-14. This looks the only place with this kind of typing info.
Line 20. is {} different from $\bot$?

Algorithm 3:

Lines 47-50. Are the =, logical tests from invariants or states changes, as in $\leftarrow$ ?

Algorithm 4:

Line 149 uses an $\equiv$. Maybe this symbol is not needed, and just make sure everywhere to use = for testing equality and $\leftarrow$ for changing state.

Page 14:

The first paragraph in this page "We define ..." looks a bit out of context.

"a client can be committed — Validity follows." -> "a client can be committed. Validity follows."

Page 15:

"delivered in the same batch, and" -> "delivered in the same batch."

Page 19:

In figures X axis, instead of 20,40,60 you could label the actual n used, like 16,32,48 ...

Page 20:

"with a drop from 28.3k rew/s" -> "with a drop from 28.3k req/s"

Page 21:

"(resp,. second)" -> "(resp., second)" or "(resp. second)"

Page 23:

"Aardvark [24] was the one the first BFT protocols" Remove the

**Expertise:**

Follow the literature for the broad topic of this paper

**Useful:**

yes

**Writing:**

Non-experts can understand the paper

---

### Official Review · Reviewer_xQ47 · 2021-06-09
**Highly detailed BFT system design utilizing parallelism**

**Decision:**

Weak accept: good paper with flaws that can be fixed in three months

**Review:**

This paper presents a BFT system that has concurrent leaders proposing request batches in order to improve the throughput via parallism. The paper makes commendable efforts in providing detailed pseudocode, correctness proofs and evaluation results. But the paper also has several issues that need to addressed in a revision.

The paper claims that no prior work has ever studied parallel leaders with duplicate prevention. I have doubts about this claim. The HoneyBadger BFT paper discussed this exact matter. It uses a common subset procedure that commits multiple proposals, and also has a discussion on avoiding duplicates. The solution is very similar to this paper: assign requests to proposers using a hash function. And I think the HoneyBadger paper credits even earlier papers for observing this challenge and solution. The authors should cite those papers and reconsider the how novel this method is. To be honest, even without those prior works, I find the idea of hash partitition straightforward.

Once you employ the hash partitioning, a client request is only assigned to one single node. This raises the natural issue  that a Byzantine node can now censor a transaction. The paper touts censorship resistance as a main feature of the system, but the solution here (reassigning buckets) is also quite straightforward and a bit underwhelming as it allows Byzantine nodes to censor a transaction for f "periods".

Given the above, I suggest the authors tone down claims about proposing certain designs or addressing certain attacks for the first time. The paper makes solid contributions by implementing simple and natural (and likely existent) ideas of random request assignment and demonstrating improved performance over existing systems. This is strong enough for a systems paper.

The paper uses vanillar PBFT as the baseline in an overly rigid fashion. While BFT is undoubtedly a seminal work, it has been more than two decades since it invention. There has been a lot of progress and new understanding since then. More notably, the PBFT view change has been improved by the Tendermint and HotStuff paper. I suggest the authors incorporate the latest progress in BFT. For a few minor points, it seems unnecessary to emphasize that PBFT has no batching. It is pretty obvious how to add batching to PBFT and PBFT-style protocols.

In terms of the writing, while it is commendable that the authors provide super detailed pseudocode and proofs, I have to say they are not very helpful to a reader. The code is too detailed and low level. Most of the steps in the code and proofs are adopted from PBFT and hence are not really useful to understanding this paper. It also goes against the idea that SRB simplifies the reasoning by building on top of a well understood system. In that spirit, it will be more helpful if the authors can pinpoint and highlight the differences to PBFT in the pseudocode and proofs.


=== Other minor comments ===

Section 3 mentions "bottlenecks" in the title but did not discuss them.

Section 4: the third paragraph makes it sound like batching and watermarking are two competing and incompatible approaches. I don't think that is what you mean there. Please clarify.

Page 5 says that the primary "reliably broadcast" its selection. It is unclear what "reliable" means here. Do you mean Bracha-style reliable broadcast?

It is mentioned that  leaving  the payload out of the hash can prevent  the client from manipulating the bucket assignment. This is not clear as the client still has control over the timestamp and may still manipulate to some degree.

Section 8 feels like a rather minor point . If some nodes do not need to commit the request, one can think of request hash as the "committed values" and the totality property naturally follows.

**Expertise:**

Actively publishing in this area

**Useful:**

yes

**Writing:**

Non-experts can understand the paper

---

### Official Review · Reviewer_BNUj · 2021-06-09
**Review for paper titled, "Solution: SRB: Scalable and Robust BFT for Decentralized Networks"**

**Decision:**

Weak reject: interesting papers with flaws, not sure if they can be fixed in three months

**Review:**

## Summary:
This paper aims to scale existing BFT consensus protocols by allowing multiple each node to act as a leader and run independent consensuses in parallel. To achieve this task, the paper proposes SRB that follows the seminal PBFT protocol and runs up to n instances of PBFT in parallel. SRB works n epochs. In each epoch, one of the replica is chosen as the primary, and the primary chooses a set of epoch leaders. Once a leader has received sufficient number of client requests, it proposes it as a batch to all the other nodes. As a byzantine client can send the same request to multiple leaders, SRB divides the client requests into buckets. Each leader only proposes requests assigned to its buckets. In the case any leader fails or acts malicious, SRB switches to the new epoch with a new primary leading the epoch and determining the next set of leaders.

## Pros:
1. This paper represents a timely problem as it is critical to design BFT protocols that are not dependent on a single primary. SRB bolsters system throughput and facilitates load balancing by allowing multiple leaders to propose requests in parallel.
2. The paper is easy to read and clearly motivates the problem at hand.
3. The paper presents a good set of experiments to evaluate the effectiveness of SRB against the presented protocol. Further, the paper includes a set of failure scenarios, which have been explored well during experimentation.

## Cons:
1. My main concern with this paper is that it completely misses the state-of-the-art protocols, such as Omada[1] and RCC[2][3], which also parallelize consensus. Hence, this is not the first paper to allow up to n nodes to run consensuses in parallel as claimed by the paper. Omada was the first paper to allow up to n nodes to propose requests in parallel. In recent years, RCC has been proposed which does the similar tasks with some improvements under failures. In fact in the common case operations, SRB can do no better than Omada and RCC as all the three protocols can allow all of their nodes to act as leaders. This is a noteworthy observation, which this paper should have highlighted and presented in the evaluation. Hence, this paper needs to add extensive comparison with Omada and RCC.
2. The paper highlights the ability of a byzantine client to send duplicate requests as a major concern for the multiple leader protocols. To resolve this challenge, the paper presents the bucket distribution scheme as a novel solution. Although this solution works, RCC handles this problem in a similar manner by dividing clients equally among the leaders. The key difference is that SRB continuously switches the buckets, while RCC switches clients across leaders only under failures. Hence, it is important to understand the effectiveness and costs of both the schemes.
3. The paper presents no discussion on request execution. For a protocol allowing multiple leaders, it is important to describe the order in which the client requests be executed. If one or more leaders fail, do the replicas continue executing requests from other replicas. Clearly, this should not be the case as there will be a safety violation. Hence, how do nodes realize the right time to execute the client requests. How much time do nodes have to wait for executing requests post failures.
4. Although SRB can recover itself from the failure of its leader nodes, it presents a blocking design where failure of one leader causes the epoch to change. During this epoch change all the leaders have to stop proposing new requests and the nodes can no longer execute requests until the system is recovered. RCC on the other hand proposes a design where failure of one leader allows other leaders to continue proposing requests as the view change is localized and there is no notion of epoch change. Hence, it is important that the paper illustrates the performance of SRB against RCC under such failures.
5. In Section 5.6, SRB adds a tricky optimization. When a failure-free execution starts observing simple failures, message delays or loss, even if a replica has sufficient number of Prepare messages, it will continue waiting until timeout for Prepare from the selected f+1 replicas to arrive. In fact, a byzantine replica may just play this as a trick on some other replica, even though nothing bad has happened to the system. As a result the system may face severe performance degradation, or worse could undergo an epoch change.
6. The paper includes a lot of code, which could come really handy if there is some accompanying explanation. In its current form, the code is hard to follow.

## Other Comments:
1. On Page 7, in the last paragraph, the paper states that conditions 5 and 8 are not met by PBFT, which is not a correct claim. Condition 6 is implicit in PBFT. PBFT does not force clients to send only one request at a time, and clients do add timestamps to each request. PBFT's OSDI'99 version does include clients signing request with digital signatures. The notion of MACs is included in PBFT's journal version, which is further expanded in Castro's thesis.
2. Section 5.4.1 needs to be re-worked with the details regarding the messages transmitted clearly specified. the paragraphs in this section are trying to borrow ideas from two different variations of the PBFT (conference and journal versions). Such variations may be difficult to understand for a reader who is not well-versed with the working of both the PBFT variants.

## Typos:
1. such such inconsistent epoch.. [Pg 8]
2. Before starting to execute participate in... --> sentence needs reformatting

## Reference
1. 	M. Eischer, T. Distler: Scalable Byzantine fault-tolerant state-machine replication on heterogeneous servers. Computing 101(2)
2.  S. Gupta, J. Hellings, M. Sadoghi: Brief Announcement: Revisiting Consensus Protocols through Wait-Free Parallelization. DISC 2019: 44:1-44:3
3. S. Gupta, J. Hellings and M. Sadoghi, RCC: Resilient Concurrent Consensus for High-Throughput Secure Transaction Processing, ICDE 2021 [arxiv]

**Expertise:**

Actively publishing in this area

**Useful:**

yes

**Writing:**

Non-experts can understand the paper

---

### Official Review · Reviewer_uTvN · 2021-06-09
**Interesting approach to a multiplexing strategy over PBFT**

**Decision:**

Weak accept: good paper with flaws that can be fixed in three months

**Review:**

The paper presents a very interesting approach to generalizing PBFT. It claims one primary contribution: higher-performance (throughput, specifically) BFT TOB, which is unaffected by request duplication attacks. The system is called SRB and there is a thorough protocol description, some proofs of correctness, and system evaluation. The evaluation is quite extensive though some details could help improve it. In general, the paper is well-written but is missing some important details and clarifications. The style and presentation is mostly clean, but the authors rarely make efforts to simplify and make the topic accessible (i.e., "Only experts can understand the paper"). In this sense, there is room for improvement before it can be considered a quality publication.

Takeaway:
I think the paper is relatively strong and very interesting, but has multiple serious flaws. It's possible that these flaws can be simply fixed by being more clear, but it's also possible some more extensive experiments would be necessary (wrt. batching, Chain's performance, and the dubious claim of "best throughput to date"). I remain optimistic that the authors can fix them within ~3 months.

# Weaknesses

- It would improve the quality of the paper if you are more precise early on (Intro) about which "state-of-the-art BFT" systems you are comparing to.

- The major novelty seems to be in preventing request duplication attacks.
    - If I understand correctly your solution, I would propose that the concept of a "duplicate request" does not make much sense in the context of an ordering service (which is what SRB is). Concretely, your interface is `BCAST(r)` which a client calls once for every distinct `r`. Duplicates make sense when there is a higher-level abstraction, e.g., commands (like in an SMR) or transactions (like in a UTXO model), and in that case you could say that two inputs to the service are duplicates. But if a client calls `BCAST(r)` twice with identical input `r` the service is agnostic to it and should treat each call as a separate request and should order the same `r` twice.
    - This clear distinction as to what is a duplicate request in your model is lacking and clarifying that would be a great improvement to your paper.

- In general, it is more valuable to show fewer evaluation results if that means
  that they are higher quality and more thoroughly explained.
    - Some questions and suggestions below...
    - I find it surprising that SRB manages to beat Chain consistently, except for the cluster environment (where Chain uses MACs, Fig.7). Is it because Chain has no batching? This may be an important detail you overlooked, so please clarify.
    - SRB implements a total-order broadcast, while some of your competitors (PBFT, from [15], for instance) do more than total-order and also have an execution engine. Is that engine disabled? Or is the overhead from passing transactions through an execution step still present in some competitors, while SRB omits that? It would be good to clarify since this is important.
    - You mention that the VMs have a 1Gbps network and I'm trying to understand how does that translate into Chain's throughput in Fig.4(a), since you pointed out that the bottleneck is probably the weakest link. A back of the envelope calculation shows that this link is using around 20000 (req/s) * 500 (bytes) = ~0.08 Gbps, which is one order of magnitude smaller than the theoretical bandwidth you have of 1Gbps.
    - If Chain's performance is indeed limited by the weakest link, then I would expect in a growing network size to see lower and lower throughput, since there would be higher chances of traversing weaker links. But Fig.4a shows roughly fixed (or even growing) throughput for "Chain-500b". Can you clarify?
  - Some other recent papers also claim "excellent performance", and some do evaluation on even larger setups (~200 nodes). I am thinking of [25] and [33] for example. It would have been good if you compared against them in an evaluation, since that would prove your case of "best throughput to date on public WAN networks". With the present evaluation, your claim is not on a very solid ground. You could just rephrase that to be accurate by saying "best throughput among the systems we evaluated"...

- Section 9.4 is impressive in showing the importance of preventing duplicate requests, though as I mentioned earlier I believe the definition of a "duplicate request" should be clarified, since this seems something specific to your system and not a concern in your competitors (PBFT, HotStuff).

- I appreciate that you took the effort to put the pseudocode of PBFT in section 6, and also that of SRB. But it would be more valuable and respectful for the readers (and reviewers) if the pseudocode was explained in text, e.g., you could weave the algorithms with your text of section 5 (and 3), and in that way the two sections could support each other consistently.

- Please clarify the difference (if any) between your signature verification sharding technique (5.6)
  and other protocols that use similar optimizations [25, 33].


# Strengths

- builds on the seminal PBFT protocol, instead of building the solution up from scratch
    - This evolutionary (vs. revolutionary) approach has nice benefits

- The buckets approach and periodic redistribution (sec.4) is very neat!
    - (Though I didn't find Fig.2 very intuitive or helpful, consider simplifying that figure.)

- I found the "emulation" of other BFT protocols very interesting.

# Questions

- Comparison with mempool approach:
    - "In more established
      decentralized systems, such as Bitcoin and Ethereum, it
      is standard to charge for the same transaction only once,
      even if it is submitted by a client more than once."
    - In such systems, the same transaction does not usually end up
      in more than a block, either because of the UTXO model or because the
      mempool implementation takes care of removing already-ordered txs, so
      charging for the same transaction does not apply, but if a transaction
      did end up included multiple times in the blockchain, it would pay fees
      multiple times.
    - It would be good to be more accurate with your description of other systems. Also, your approach has the flavor of a mempool (line 59, alg 4), but it is sharded.

- " While many successor BFT protocols eliminated watermarks in favor of batching (e.g, [13, 16, 39])"
    - Is watermarking the same as pipelining? In PBFT, watermarking functions as a pipelining optimization. Please clarify.
    - I looked at the references and could not find evidence that watermarks are eliminated in those protocols. While they don't mention water marking explicitly, some avoid watermarking ([16]) while some do use pipelining, e.g., [13] (which I understand to be similar to watermarking). And it's confusing to say that watermarking would be eliminated in favor of batching, since these two techniques are not mutually exclusive.

- You cite the journal version of PBFT (ACM TOCS'02) and state in Table 2 that this protocol has "no" batching. I found that strange so I looked at the paper again, and found in 6.1 that it describes request batching. So please double-check this statement and try to be more accurate.

- I see how your solution generalizes the idea of primary, to permit an arbitrary number of primaries. Does your solution resemble the following thought experiment, and how is it different if so? Suppose we simply run X concurrent instances of PBFT among the same N nodes, and each instance `i` of PBFT writes its client requests in a specific log (no execution of client requests, just log them in a database/file), then a separate process would merge these log files deterministically, in the same manner at all N nodes?

- "SRB implements all of these and is thus robust in the Aardvark sense."
    - Does SRB indeed implement separate NICs? That was not clear.

- I am not contesting your claim that the bottleneck in BFT TOB is the leader's network (though some other papers claim differently), just to note that some systems take alternative approaches to relieving this bottleneck. One example is to avoid the broadcast primitive which the leader typically uses, and instead let a gossip overlay load-balance the pre-prepare step (and all steps in fact). Some new systems are starting to build on gossip (beside the usual suspect Bitcoin, there is RapidChain and Tendermint that come to mind, the latter being closer to your model). I suspect a highly-efficient gossip overlay is a simpler approach to relieving the leader bottleneck, and I would be curious to know how would this approach compare to your multiplexing strategy.

# Potential improvements in writing and clarity

- section 3 title is "PBFT and its Bottlenecks" but there is a single bottleneck, at the primary node at step pre-prepare, correct? I was wondering if there are other bottlenecks and I missed some.

- While I appreciate the evolutionary approach, it would be nice to be precise about what exactly are you changing from PBFT (e.g., Section5.6 mention common-case changes, but your delta to PBFT is quite substantial). It does not help that you include the bulk of the pseudocode in section 6 and there is no apparent common-case reuse (despite claiming that you simply generalize PBFT and only affect liveness).


# Further ideas

- make it clear, early on, that the solution is deterministic (as opposed to randomized solutions, e.g., HoneyBadger, Algorand,...)
- "Gracious" means kind and warm (see dictionary). I think it may be more precise to use "graceful".
- Probably a typo, please fix: "This paper presents SRB (or, simply, SRB)"
- Please take effort to consolidate the writing and provide intuitions where possible (not sure table 1 is necessary, the pseudocode can be reduce and focused on what is important / delta to PBFT, LTO section is out of place)

**Expertise:**

Actively publishing in this area

**Useful:**

yes

**Writing:**

Only experts can understand the paper

---

### Author Response · Authors · 2021-09-17
**The revised version is ready for inspection**

 The revised version is ready for inspection

---

> ### Comment · Reviewer_BNUj · 2021-11-07
> **Meta-Review by Reviewer BNUj**
>
> Dear authors,
>
> All the reviewers agree that the authors have done a good job in revising the paper. However, some key points remain unanswered, and we expect the authors to address these comments in the final draft of their paper. As a result, we are moving this paper to the shepherding stage. Following are the points that need to be addressed:
>
> - A baseline performance comparison between SRB and parallel PBFT with a simple workload where client does not submit any duplicates.
> - SRB claims that prior work RCC suffers from request duplication and request censoring attacks. However, the paper provides no information how RCC suffers from these attacks. Like SRB, RCC also deterministically assigns each client to only one specific primary. Authors should provide specific examples which show stated attacks on RCC.
> - The paper states that RCC faces request censoring attack, which SRB is able to avoid. In the case of RCC, even though each client sends a request to only one primary, each client on timeout has a defined mechanism of contacting other replicas and request a primary switch. Hence, the authors should illustrate examples that prove the stated attack on RCC.
>
> We are looking forward to your final version.

---

> > ### Author Response · Authors · 2021-11-18
> > **Towards a final version (pt1)**
> >
> > Dear reviewers,
> >
> > Thank you again for your valuable feedback.
> > We would like to briefly discuss your comments so we can converge to a final version of our paper.
> >
> >  > A baseline performance comparison between SRB and parallel PBFT with a simple workload where client does not submit any duplicates.
> >
> > Our SRB (noRotation) implementation, compared to SRB in Figure 8 (referred in Sec. 9.3) with a simple workload without duplicate requests, is the baseline you are describing: Parallel PBFT instances, sharing only a common checkpoint mechanism and watermark window.
> > We can better clarify this in our final version.
> >
> > We should also clarify that this is not a live protocol, unless the client can resubmit their request (potentially after a timeout, substantially increasing latency) to at least f other nodes - similarly to BFT-Mencius or RCC.
> > Using timeouts between request resubmissions would optimize throughput only in a fault-free synchronous execution. Crucially, such an approach is vulnerable not only to faulty clients (the number of which we do not bound by assumption), but also to asynchronous client network connections. Therefore, in Fig. 9, we only further explore the case where the client immediately submits their requests to f+1 nodes.
> >
> >
> > > SRB claims that prior work RCC suffers from request duplication and request censoring attacks. However, the paper provides no information how RCC suffers from these attacks. Like SRB, RCC also deterministically assigns each client to only one specific primary. Authors should provide specific examples which show stated attacks on RCC.
> > We agree that it was inaccurate to group RCC with other protocols that do not address request duplication, since, indeed, RCC introduces a mechanism to mitigate this issue. However, we find the proposed mechanism insufficient in case of malicious clients. In fact, RCC suffers multiple performance attacks due to malicious clients. We suggest expanding related work according to the following arguments.
> >
> > * **Uneven client load distribution:**
> > In RCC, clients can freely choose the primary node whose instance they will be assigned to. Therefore, all malicious clients can submit their requests to the same primary, effectively converting RCC to a single-leader protocol.
> > Importantly, a malicious client requesting to switch instances cannot be distinguished from a slow one whose request execution has been delayed due to network asynchrony. Analogously, even honest clients might end up being assigned to the same instance only due to asynchrony.
> > In contrast, SRB hashing approach along with the client watermark window aims at more uniform load distribution.
> > *  **Duplicate proposals:**
> > Using consensus to agree to which instance a client is assigned eventually guarantees that a client will be assigned to one instance. However, in an asynchronous network we can construct an execution where a client sends a request to some instance and then asks to switch instance such that both instances *propose* the request, wasting bandwidth on the same request.
> > This is possible because (1) nothing prevents an instance from proposing a request that has already been accepted, and (2), even if RCC was hardened with a mechanism to prevent (1), an adversary controlling the message delivery schedule can delay the acceptance of a proposal long enough so that some other primary proposes the same request.
> > This can be extended to more than two instances proposing the same request.
> > Again, a malicious client cannot be distinguished from a client whose messages are delayed by the network.
> > SRB completely prevents multiple proposals of the same request.
> > * **Forced re-transmissions:**
> > RCC prevents censoring by having non-primary replicas re-transmitting a request that is not proposed fast enough by the primary. This mechanism can be exploited by a malicious client who sends their requests to all replicas except for the primary, causing non-primary replicas to waste bandwidth on retransmissions and the primary to waste bandwidth on receiving the retransmissions O(n) times. In practice, though, this re-transmission cost could be minimized with simple optimizations.
> > * **Instance synchronization vs wait-free design:**
> > RCC uses a parameter σ to prevent malicious clients from subscribing requests to multiple instances and, also, to prevent a primary from acting as a straggler. However, σ is underspecified (it is not clear what “being behind” means exactly) and we suspect that synchronizing replicas so that they are never more than σ  rounds “behind” each other violates RCC’s wait free design (intuitively, never preventing some instance from making progress (wait freedom) seems to imply that this instance can get arbitrarily “ahead” of other instances).
> >
> > We should also mention that SRB preprint precedes RCC and is already cited in RCC paper.
> >
> > (continues in pt2)

---

> > > ### Comment · Reviewer_BNUj · 2021-11-21
> > > **SRB and RCC discussion**
> > >
> > > The response by the authors helps to bring some interesting points into view. However, here is my take and I am looking forward to the authors reply to the same.
> > >
> > > * **Uneven Client load distribution:** I fear that there is a wrong conclusion here. The RCC paper does not state that clients can choose which primary they want to. From what I read, it seems like each client is assigned to some primary. Further, RCC like SRB aims to load balance, which implies that clients will be split across primaries. Even in the case that all malicious clients are assigned to same primary, I am unable to understand how does that make RCC a single-primary protocol because the remaining good clients will be split across rest of the n-1 primaries. I agree that due to asynchrony attacks several clients may not be able to reach their respective primaries and could request instance switch, but RCC only allows clients to switch after consensus between all instances. So I do not see a case when uneven client load distribution happens.
> > >
> > > * **Duplicate proposals:** I am unsure how the proposed attack will take place. First, like SRB, RCC also makes use of primary-backup protocols like PBFT. In these protocols, every replica checks if the client request has a new timestamp. So, a replica will check timestamps prior to proposing any request. Further, client can only switch primaries only after all the replicas have reached consensus. So, replicas are aware of a clients old instance and new instance. Moreover, a new instance only proposes requests after some $\sigma$ rounds.
> > >
> > > * **Forced Re-transmission:** This attack which authors have explained, unless I am mistaken, is applicable to several BFT protocols, such as PBFT, SBFT, and so on, including partly to SRB. SRB also states that the client will re-transmit request to all the replicas if it does not receive response. A malicious client can always exploit this and send the request to all the replicas in the first place. I agree that SRB will not cause forced re-transmission to the specific leader, but in worse case it will cause delay of f+1 bucket switches as first f of the leaders receiving request could be byzantine. This means good clients in SRB will have high latencies, which is not faced by RCC. So, the real tradeoff is between bandwidth wastage versus latencies, something important to note in the paper.
> > >
> > > * **Instance synchronization:** SRB states that even if one of the leader is byzantine or sufficient number of replicas timeout, an epoch change will occur. This epoch change is equivalent to view-change protocol by PBFT. This implies that SRB will have down periods during epoch change when system throughput will be zero. There is nothing wrong with this as it is a standard design. In comparison, RCC avoids this design as it continuously orders request till there are good primaries. Regarding $\sigma$, I agree it suggests some synchronization, but it seems like a deployment parameter, like other deployment parameters, such as bucket switching period in SRB.
> > >
> > > The claim of which paper precedes which paper seems orthogonal to the technical details of each paper; the earlier version of RCC was published in DISC 2019.

---

> > ### Author Response · Authors · 2021-11-18
> > **Towards a final version (pt2)**
> >
> > (continues from pt1)
> >
> > > The paper states that RCC faces request censoring attack, which SRB is able to avoid. In the case of RCC, even though each client sends a request to only one primary, each client on timeout has a defined mechanism of contacting other replicas and request a primary switch. Hence, the authors should illustrate examples that prove the stated attack on RCC.
> >
> > We did not intend to claim that RCC suffers from censoring attacks.
> > Could you, please, point us to the source of this misunderstanding in the manuscript? We will do our best to clarify it.
> >
> > We can provide you with a new manuscript that incorporates the proposed changes (with the changes highlighted).
> > How can we submit this intermediate document to you? This does not seem currently possible using OpenReview.

---

> > > ### Comment · Reviewer_xQ47 · 2021-11-18
> > > **Duplicate elimination**
> > >
> > > Dear authors,
> > > I will let other reviewers follow up on RCC. But I have a related question: your revision states that the duplicate elimination technique of HoneyBadger BFT does not work well. Can you elaborate on its limitation and why your scheme resolves it? Thanks

---

> > > > ### Author Response · Authors · 2021-11-19
> > > > **HoneyBadger BFT**
> > > >
> > > > In the revised submission we added a paragraph in Sec 1 (Introduction) to clarify this.
> > > > I copy the paragraph here to facilitate the discussion.
> > > >
> > > > ```
> > > > The second case refers to the approach taken in Honeybadger.
> > > > Notably, the authors realize the importance of duplicate elimination.
> > > > Their random sampling approach would result to no duplicates on expectation
> > > > with a sufficiently large pending queue.
> > > > However, in practice, unless the system is deep in saturation,
> > > > the pending queue does not contain significantly more requests than the next batch.
> > > > Therefore, if the pending queue of multiple leaders contains duplicate requests,
> > > > the leaders will include them in their respective batches.
> > > > ```
> > > > We will move this paragraph to related work in the final version and leave a summary of it in the Intro.
> > > >
> > > > Please let us know if we need to further clarify.

---

> > > > > ### Comment · Reviewer_xQ47 · 2021-11-19
> > > > > **HoneyBadger BFT duplicate**
> > > > >
> > > > > Thanks. I saw this new paragraph in the paper but did not follow it. Why are we comparing the pending queue with the next batch?

---

> > ### Author Response · Authors · 2021-11-24
> > **SRB shepherding discussion (pt1)**
> >
> > Dear reviewers,
> >
> > Thank you again for the feedback.
> > Below (pt2 of this comment) we address the latest comments on HoneyBadger and RCC.
> > We hope we can soon converge to a fair and understandable related work section.
> >
> > At this point we would also like to raise our concerns about the progress of the shepherding process  and the lack of concrete guidance we perceive. As, according to the decision response, we should submit the final version by December 7th, in the interest of time, we would like to ask the shepherd to help us converge to a concrete, final list of modifications of our manuscript.
> >
> > At any moment, once OpenReview allows it, we can submit the latest version of our manuscript which has well annotated changes according to the discussion so far.
> >
> > (continues in pt2)

---

> > > ### Comment · Program_Chairs · 2021-11-29
> > > **You should be able to post your revised version now**
> > >
> > > Dear authors,
> > >
> > > I just modified the OpenReview settings such that you can now submit your updated version with annotated changes. (Please let me know asap if that should still not work...).
> > >
> > > We will make sure that, shall there be still open points to be corrected, you will be provided with a clear list of expected modifications.

---

> > ### Author Response · Authors · 2021-11-24
> > **SRB shepherding discussion (pt2)**
> >
> > (continues from pt1)
> >
> > ## HoneyBadger duplication and its Pending queue
> >
> > The pending queue is a node’s buffer of received requests that have not yet been proposed. When a node constructs a new proposal, it uses requests from this buffer. If two HoneyBadger nodes concurrently propose a batch, it may contain duplicate requests, as those might be present in both node’s pending queues. HoneyBadger does not prevent this from happening, targeting “mostly disjoint sets of transactions” in its proposals. We will clarify this in the next draft.
> >
> > ## RCC
> >
> > While we strive to get the details about RCC right according to the published version of the paper, we do not wish this thread to derail to a non-conclusive discussion around RCC. We hope we can conclude that both approaches have their advantages.
> > The RCC design reduces the impact of failure recovery on throughput, while SRB reduces the impact of malicious (or slow) clients on throughput.
> >
> > Below we address the specific comments in detail.
> >
> > ### Uneven client load distribution
> >
> > The RCC paper states that a client “can request to be reassigned to another instance“ and that this reassignment is handled by a consensus protocol that “reaches agreement” on such requests. It is not stated, however, that such a request can be rejected and under which conditions this happens. We thus interpreted it as the protocol accepting such requests once consensus on their reception is reached.
> >
> > We do not claim that RCC is a single-leader protocol, but that it can end up behaving like one in presence of asynchrony or malicious clients. Note that the number of malicious clients is not bounded by assumption and thus most, even all clients can be malicious.
> > Clients switching instances only after the system reaches consensus on the switch (assuming “replicas” were meant instead of “instances” in the one-before-last sentence) does not by itself guarantee balanced client distribution, unless additional protocol logic is applied and reassignment requests can be rejected.
> >
> > ### Duplicate Proposals, Instance Synchronization
> >
> > Following Section III.E of the RCC paper, let us assume the following execution:
> >
> > A client c submits a request r to instance I_i and the leader of that instance *proposes r*. Before r is committed, however, c sends m = SwitchInstance(c, j) to all replicas, the replicas agree on m and c submits r to instance I_j. I_j *proposes r* after sufficiently many rounds (3σ or more) pass.
> >
> > The RCC paper does not mention a timestamping mechanism, but we assume it inherits PBFT’s (logical) timestamping mechanism: request with timestamp t will not be accepted after it has been committed.
> >
> > Indeed, we are now convinced that if R_j (leader of I_j) is aware of the switch of client c *and* about the outcome of the agreement on r before proposing r again, request duplication can be prevented in RCC. However, R_j cannot be aware of the outcome of the agreement on r within σ rounds unless it either (1)  R_j blocks and waits for it, or (2) we rely on stronger than eventual synchrony assumptions.
> > Thus, it seems that one cannot have both wait-freedom (in the RCC sense) and prevent duplication at the same time in RCC.
> >
> > ### Forced Re-transmission
> >
> > Indeed the request retransmission is a mechanism already present in other protocols, including PBFT.
> > We designed SRB with that in mind, so such an attack is not possible.
> > In particular, SRB does not rely on re-transmission by the *nodes* to guarantee liveness.
> > It relies on the clients submitting the request to enough nodes and on the bucket re-assignment mechanism.
> > There is a tradeoff to this approach.
> >
> > * On the one hand, SRB nodes do not waste bandwidth on request re-transmissions.
> > * On the other hand, unlike PBFT, RCC, etc.,  SRB nodes will not **immediately**¹ suspect a node that censors requests, and therefore client requests that are censored will have an increased latency because they need to wait for their bucket to be re-assigned. We evaluated this latency in the censoring experiment and the results are found in Figure 11 (Section 9.5).
> >
> > In Figures 11a, 11b we observe that this latency remains in the order of seconds.
> >
> > ¹**immediately** hides some timeout that needs to expire before the re-transmission that could also be in the order of seconds.
> > We, therefore, decided to favor throughput by preventing re-transmissions, since the penalty on latency is comparable to the timeout.
> >
> > Since our design choice to not use the re-transmission mechanism is not justified in our manuscript, we can add the argumentation above.
> >
> > Long sequences of leader failures are an orthogonal problem from which any timeout-based leader failure detection suffers. SRB’s up to f view changes in a row are, from the point of view of request latency, equivalent to up to f instance changes of RCC.
> > It is true, however, that SRB’s throughput suffers significantly more during view changes.

---

> > ### Author Response · Authors · 2021-12-01
> > **Revised version**
> >
> >  We uploaded the latest version of our manuscript with well annotated changes in sections 1,9,10 according to the discussion.

---

### Meta-Review · Area_Chair_z9se · 2021-06-22

**Recommendation:** Revise
**Confidence:** 4

**Metareview:**

The reviewers agreed that the paper presents an interesting generalization of PBFT and unlike other protocols it manages to be as close as possible to PBFT, which is the most widely deployed protocol. As a result, it has good chances of adoption. However, there are multiple open questions pertaining to the motivation of some parts (e.g., the usefulness of the transaction deduplication mechanism) and the fair addressing of related work (see reviews). As a result, we would like to see a revised version addressing the following key points:


1. Authors should clarify what the term duplicate requests implies in their system. As noted by a reviewer, this term has an impact in terms of commands or UTXO model, but does not affect the functioning of an ordering service.
2. Authors should theoretically and experimentally evaluate the effects of running n-concurrent instances of PBFT at all the nodes, and combining the results. How would such a system differ from SRB.
3. A key motivation for the proposed protocol is that the leader has to perform more tasks than other nodes (broadcasting the request to all the nodes). Can state-of-the-art gossip primitives improve the performance of leader nodes and in extension the performance of existing protocols.
4. SRB compares its performance against vanilla PBFT implementation. Several recent works have presented improved PBFT implementations that employ techniques such as pipelining, batching, and watermarking. It is important to note the performance of SRB against such a state-of-the-art PBFT implementation.
5. The paper needs to revisit its claim as the first protocol to parallelize BFT consensus. For instance, HoneyBadger also partitions proposals among different proposers, and it is important to understand how SRB's design differs against such a protocol. Further, recent protocols such as Omada and RCC provide algorithms that allow all the nodes to act as leaders. The paper should highlight how it theoretically and experimentally differs from these protocols.
6. The paper needs to explain how SRB executes different client requests. Does the failure of one leader forces the execution of other requests to pause. If not, how does the protocol guarantee safety?
7. SRB requires nodes to switch to a new epoch if any leader fails. This blocks the system from processing newer requests until the start of the next epoch. In comparison, RCC allows the system to continue processing requests even if some leaders fail. The paper should discuss this scenario and suggest ways in which it can make SRB non-blocking.

---

### Decision · Program_Chairs · 2021-06-16

**Decision:**

Accept

**Comment:**

Hi authors,

Congratulations! The reviewers have converged on the decision to accept your paper with shepherding. The area chair and shepherd will explain the changes that they want to see in the final version of the paper. We have determined this can be done within the one month available for authors.

Please note, your one month will start from the date at which the area chair and shepherd lay out the changes required in the final version. Congratulations again, and thank you for working with the reviewers on this paper!

Editor-in-Chief, JSys